# Specific pharmacological and $G_{i/o}$ protein responses of some native GPCRs in neurons

Chanjuan Xu [1,2,5], Yiwei Zhou[1,4,5], Yuxuan Liu[1,5], Li Lin[1], Peng Liu[1], Xiaomei Wang[1], Zhengyuan Xu[1], Jean-Philippe Pin [3] ✉, Philippe Rondard [3] ✉ & Jianfeng Liu [1,2] ✉

G protein-coupled receptors (GPCRs) constitute the largest family of membrane proteins and are important drug targets. The discovery of drugs targeting these receptors and their G protein signaling properties are based on assays mainly performed with modified receptors expressed in heterologous cells. However, GPCR responses may differ in their native environment. Here, by using highly sensitive $G_{i/o}$ sensors, we reveal specific properties of $G_{i/o}$ protein-mediated responses triggered by $GABA_B$, $\alpha_2$ adrenergic and cannabinoid CB1 receptors in primary neurons, different from those in heterologous cells. These include different profiles in the $G_{i/o}$ protein subtypes-mediated responses, and differences in the potencies of some ligands even at similar receptor expression levels. Altogether, our results show the importance of using biosensors compatible with primary cells for evaluating the activities of endogenous GPCRs in their native environment.

G protein-coupled receptors (GPCRs) are ubiquitously expressed in every cell type, and constitute the largest family of membrane proteins[1]. They participate in the regulation of a large variety of physiological functions[2]. Their dysregulation or malfunction can be the cause of numerous diseases[3]. Accordingly, controlling their activity with selective drugs can have multiple therapeutic effects, as illustrated by the large proportion of the clinical drugs targeting GPCRs[4].

Most of the physiological functions of GPCRs are mediated through the coupling of G proteins[2]. Eventually, GPCRs are able to activate several G protein subtypes, such as $G_{i/o}$, $G_s$, $G_q$, and $G_{12/13}$ family[5–8], but this may vary depending on the use of ligands, a phenomenon called ligand biased effect[9,10], or/and the cellular environment known as system bias[11]. This concept of functional selectivity[11], that combined effect of ligand and system bias, has been largely used in the characterization of ligands[12–14], but the signalling properties and identification of potential drug candidates targeting GPCRs are performed in heterologous cells. Since the cellular environment is critical, the analysis of G protein response profile of endogenous receptors in

their native environment is therefore of much interest to validate drug candidates. Indeed, it becomes clear that many interacting proteins, post-translational modifications, localization in specific compartments, lipid membrane and ion environments largely influence GPCR signalling properties[15,16]. It is thus essential to examine the effects of drugs on the signalling properties of GPCRs in their native environments.

Functional analysis of endogenous GPCRs in their native environment is usually performed through the measurement of second messengers, such as cAMP, inositol phosphate or $Ca^{2+}$[17], or ion channel regulation[2,18]. Due to receptor reserves and signal amplification, such assays are not precise for estimating drug activity, potency and efficacy[9]. G proteins are first effectors of GPCRs, such that measuring their activation is less prone to signal amplification[18]. GTPγS binding assay can detect G protein activation in native cells or tissues[19,20]. However it requires the preparation of membranes and these assays are not specific for one G protein subtype, unless an immunoprecipitation of the G protein is added[21]. Nowadays, much hope in studying

[1]Cellular Signaling Laboratory, Key Laboratory of Molecular Biophysics of Ministry of Education, College of Life Science and Technology, Huazhong University of Science and Technology, Wuhan, Hubei, China. [2]Bioland Laboratory, Guangzhou Regenerative Medicine and Health Guangdong Laboratory, 510005 Guangzhou, China. [3]Institut de Génomique Fonctionnelle (IGF), Université de Montpellier, CNRS, INSERM, 34094 Montpellier, France. [4]Present address: Kindstar Global Precision Medicine Institute, Wuhan, China. [5]These authors contributed equally: Chanjuan Xu, Yiwei Zhou, and Yuxuan Liu. ✉e-mail: jean-philippe.pin@igf.cnrs.fr; philippe.rondard@igf.cnrs.fr; jfliu@mail.hust.edu.cn

native GPCRs is based on the development of specific optical bio-sensors that must be sensitive enough to detect the activity of native receptors[22] usually expressed at very low levels, but most of them are still not compatible with these endogenously expressed receptors[13,23–26].

In the central nervous system, GPCRs are essential regulators of synaptic transmission, acting at both pre- and post-synaptic levels, as well as in glial cells[27]. More than half of neuronal GPCRs can couple to one or more of the various $G_{i/o}$ protein subtypes[5], but their expression profiles vary in different native tissues or cell types and specific cellular function depends on distinct $G_{i/o}$ protein subtypes[12,28,29]. It is therefore essential to determine which G protein can be activated by a GPCR upon stimulation with various agonists in its native environment.

In the present study, we describe BRET-based sensors for each $G_{i/o}$ protein subtypes that can be used to study endogenous GPCRs in living neurons. Our study focuses on three types of GPCRs that play important roles in modulating synaptic activity, the $GABA_B$[30], $\alpha_2$ adrenergic[31,32] and cannabinoid CB1 receptors[33]. We could detect the activation of $G_{i/o}$ proteins from a small number of neurons and monitor both the kinetics and dose-response of the effect mediated by various compounds including agonists, antagonists and positive allosteric modulators, in different types of neurons. Our data reveal differences in the profile of $G_{i/o}$ protein subtypes mediated responses and agonist potencies between these recombinant and native receptors in HEK293 cells and neurons, respectively. In addition, our results show a major difference in $G_{i1}$ versus $G_{oA}$ protein activation induced by a CB1 receptor agonist in neurons, but not in HEK293 cells. Finally, different composition of $G\gamma$ subunits in the neurons can also lead to specific $G_{i/o}$ protein responses. Altogether, our results reveal the importance of evaluating GPCR activities in their native environment and highlight the need for sensitive biosensors compatible with native receptors in their natural environment.

## Results

### $G_{i/o}$ protein sensors for endogenous GPCRs in neurons

To measure the rearrangement or dissociation of the $G_{i/o}$ proteins upon receptor activation in live cells and neurons, we used a series of BRET-based biosensors. They are based on the use of a luciferase as an energy donor inserted in the $G\alpha$ subunit and the fluorescent protein Venus as an energy acceptor attached to the $G\gamma$ subunit, as reported for previous BRET- and FRET-based G protein sensors[23–25]. To monitor these sensors in conditions close to the physiological ones, in our approach, the cells were cotransfected only with the luciferase-tagged $G\alpha$ subunit and Venus-tagged $G\gamma$ subunit ($^{Venus}G\gamma$), while the endogenous $G\beta$ subunits were used (Fig. 1a). In these experiments, the amount of $G\beta\gamma$ complexes that can produce a BRET signal with $G\alpha$ is expected to be limited by the endogenous level of $G\beta$ subunits (Supplementary Fig. 1a) that are able to form a complex with the $^{Venus}G\gamma$. We used mostly the mouse cerebellum granule neurons (CGNs) that constitute the largest homogenous neuronal population of the mammalian brain[34].

Our sensors rely on the use of a small luciferase, the *Oplophorus* nanoluciferase (Nluc; 19 kDa) that produces a more intense and sustained luminescence signal than other commonly used luciferases from Renilla luciferase (Rluc and Rluc8; 36 kDa)[35]. Nluc was inserted in the helical domain of different $G\alpha_{i/o}$ subunits similarly to previously reported G protein BRET sensors with Rluc[23,24] and Nluc[36], and the constructs were named $G\alpha^{Nluc}$ (Fig. 1b). The Nluc was inserted after residue 91 (for $G\alpha_{i1}$, $G\alpha_{i2}$, $G\alpha_{i3}$, $G\alpha_{oA}$ and $G\alpha_{oB}$), or 113 (for $G\alpha_z$) (Fig. 1c). Of note, $G\alpha_{oA}$ has 94% sequence identity with $G\alpha_{oB}$ and around 70% with $G\alpha_{i1}$, $G\alpha_{i2}$ and $G\alpha_{i3}$[37]. To facilitate the insertion of Nluc, a short and flexible linker sequence (SGGGGS) was added at the N- and C-terminal ends of Nluc. For BRET signal measurement, Venus was fused to the N-terminus of $G\gamma_2$ subunit with the amino acid sequence KLGT serving as a linker, and the construct is named $^{Venus}G\gamma_2$ (Fig. 1b). To show the

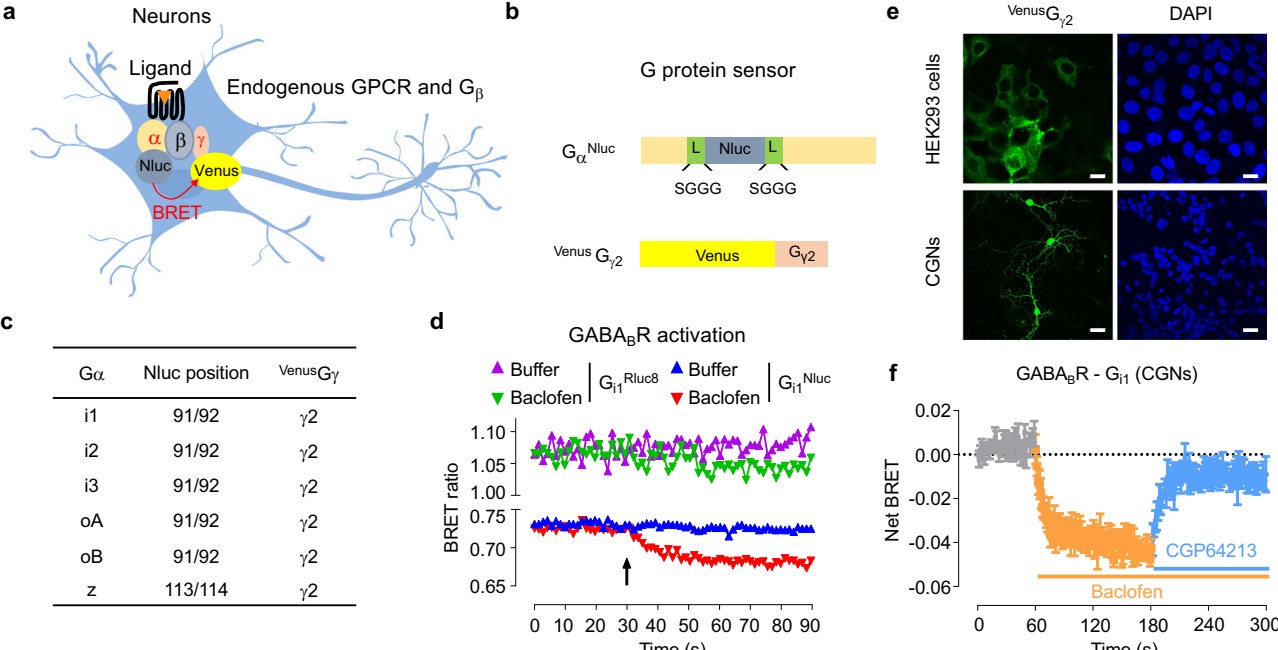

**Fig. 1 | Design and validation of the $G_{i/o}$ protein sensors in neurons. a** Scheme of the BRET-based sensors to detect the activity of the endogenous GPCR in neurons. **b** Scheme of the $G\alpha_{i/o}^{Nluc}$ and $^{Venus}G\gamma_2$ constructs co-transfected in the neurons. Sequence of the linker (L) before and after Nluc was indicated. $G\gamma$ was fused with Venus in the N-terminal. **c** Composition of G protein sensors made of the different $G\alpha_{i/o}^{Nluc}$ and $^{Venus}G\gamma_2$. **d** Kinetics of the BRET signal between the indicated $G\alpha$ constructs ($G\alpha_{i1}^{Nluc}$ or $G\alpha_{i1}^{Rluc8}$) and $^{Venus}G\gamma_2$ in CGNs. Buffer or baclofen (100 µM) were injected at the indicated time (arrow). The data are representative of BRET ratios

from five independent experiments. **e** Representative fluorescent images of Venus and DAPI in CGNs transfected with $G\alpha_{oA}^{Nluc}$ and $^{Venus}G\gamma_2$ or in HEK293 cells transfected with $G\alpha_{oA}^{Nluc}$, $G\beta_1$ and $^{Venus}G\gamma_2$ from three independent experiments. Scale bar: 20 µm. **f** Kinetics of the BRET signal between $G\alpha_{i1}^{Nluc}$ and $^{Venus}G\gamma_2$ in CGNs. Buffer or baclofen (100 µM) or competitive antagonist CGP64213 (10 µM) were injected at the indicated time. The data are representative of the mean ± SEM of BRET ratios from three independent experiments. The raw data and p-values are available in source data provided as a Source Data file.

advantage of using Nluc over Rluc8 in measuring agonist-induced BRET change in neurons, Rluc8 was inserted at the same position as Nluc in $G\alpha_{i1}$ (named $G\alpha_{i1}{}^{Rluc8}$) and $G\alpha_{oA}$ (named $G\alpha_{oA}{}^{Rluc8}$). These $G\alpha_{i1}{}^{Rluc8}$ and $G\alpha_{oA}{}^{Rluc8}$ constructs are highly similar to the constructs reported for the Gα protein of TRUPATH BRET sensors (Supplementary Fig. 1b). The $G\alpha_{oA}{}^{Nluc}$ construct showed much higher luminescence intensity at emission 480 nm compared with the subunits $G\alpha^{Rluc8}$ in CGNs, for the same amount of cDNA co-transfected with $^{Venus}G\gamma_2$ (Supplementary Fig. 1c). In addition, the basal BRET ratio of the constructs $G\alpha_{i1}{}^{Nluc}$ and $G\alpha_{oA}{}^{Nluc}$ co-transfected with $^{Venus}G\gamma_2$ was also more stable over time, compared to the sensors that used subunits $G\alpha^{Rluc8}$ (Fig. 1d and Supplementary Fig. 1d). We also verified that the co-expression of the constructs $G\alpha^{Nluc}$ + $^{Venus}G\gamma_2$ for the different Gα subunits did not change the expression level of endogenous Gβ subunits (Supplementary Fig. 1e). Altogether, the G protein Nluc-biosensors are expected to produce a much improved signal-to-noise ratio for BRET measurement in neurons.

We then validated our different $G_{i/o}$ protein sensors with the $GABA_B$ receptor as a prototype $G_{i/o}$-coupled receptor, which is abundantly expressed in many types of neurons[38], including in CGNs[39]. Interestingly, despite a low percentage of transfected neurons compared with HEK293 cells (Fig. 1e), the $GABA_B$ receptor specific agonist baclofen largely decreased the BRET signal with the sensor $G\alpha_{i1}{}^{Nluc}$ + $^{Venus}G\gamma_2$ (Fig. 1d, Supplementary Fig. 2a) and $G\alpha_{oA}{}^{Nluc}$ + $^{Venus}G\gamma_2$ (Supplementary Fig. 1d, 2b), while the change was barely detectable with $G\alpha_{i1}{}^{Rluc8}$ + $^{Venus}G\gamma_2$ (Fig. 1d) and $G\alpha_{oA}{}^{Rluc8}$ + $^{Venus}G\gamma_2$ (Supplementary Fig. 1d). In kinetics experiments, baclofen induced a rapid and strong decrease of BRET signal in CGNs, that was reversed by the $GABA_B$ receptor competitive antagonist CGP64213 for both the $G_{i1}$ (Fig. 1f) and $G_{oA}$ (Supplementary Fig. 2c) sensors. As a control, no change of BRET signal was measured when $G\alpha_{i1}{}^{Nluc}$ was expressed in the absence of $^{Venus}G\gamma_2$ or with Venus (Supplementary Fig. 2d). Similarly, no change of BRET signal was measured with the dominant negative mutants $G\alpha_{i1}{}^{Nluc}$-S47A[40] or $G\alpha_{i1}{}^{Nluc}$-G202T[41] (Supplementary Fig. 2e) when they are co-expressed with $^{Venus}G\gamma_2$. This is consistent with the unability of these mutants to exchange their GDP for GTP. We also verified that the co-expression of Gβ with the Gα and $G\gamma_2$ was not required for a large BRET change signal induced by baclofen (Supplementary Fig. 2f). Altogether, these results validated the use of the $G\alpha_{i1}{}^{Nluc}$ and $G\alpha_{oA}{}^{Nluc}$ -based BRET sensors in neurons.

These $G_{i/o}$ protein Nluc sensors were highly sensitive even though only about 2% of CGNs expressed the sensor as measured by the fluorescence of the transfected $^{Venus}G_{\gamma2}$. This low level of transfection is consistent with previous data reported with CGNs[42], even though any transfected cell is expected to overexpress the sensors thanks to the strong promotor. The transfected neurons displayed normal morphology with dendrites and branches (Fig. 1e). It indicates that the $G\alpha_{i/o}$ protein Nluc-sensors were sensitive enough to monitor endogenous GPCR activation in a small number of neurons. In addition, they are compatible with measurements in 96-well plates. The amount of cDNA to be transfected in CGNs was optimized for each sensor (Supplementary Fig. 2g). Finally, our sensors are also compatible with the measurement of the activation of endogenous GPCRs expressed in the HEK293 cells such as the lysophosphatidic acid receptor (Supplementary Fig. 3a). In these experiments, the $G_{i2}$ showed a weak response than those generated by the other $G_{i/o}$ sensors (Supplementary Fig. 3a), and this is unrelated to differences in expression levels of these sensors (Supplementary Fig. 3b).

We then compared the agonist-induced BRET signals obtained with the same $G\alpha_{i/o}$ sensor in the two different cell types, CGNs and HEK293 cells (Fig. 2a). Conditions were optimized such that the $GABA_B$ receptor expression level in HEK293 cells was similar to that in CGNs (Fig. 2b). Since the basal BRET signal in the absence of agonist can be different between the G protein sensors (Supplementary Fig. 3c), the signal was expressed as the agonist-induced change in BRET ratio

expressed as percentage of the basal signal. The results showed that in CGNs, the BRET changes induced by baclofen are significantly different from those measured in HEK293 cells except for the two $G_o$ sensors (Fig. 2a). Indeed, a larger BRET signal is measured in CGNs with $G_{i1}$ and $G_{i3}$, and a large signal is observed with Gz in HEK293 cells while absent in CGNs. The differences observed are related neither to the expression levels of the $G\alpha^{Nluc}$ sensor components (Fig. 2c) nor to the endogenous $G_{i/o}$ mRNA expression levels (Fig. 2d). Though lower signal was observed in $G_{i2}$ in CGNs, the difference of baclofen response can be detected in dose-dependent manner (Supplementary Fig. 3d). Similar $pEC_{50}$ values were measured in three $G_i$ subtype sensors and two $G_o$ subtype sensors in CGNs upon baclofen activation (Supplementary Fig. 3d). Altogether, the data showed different abilities of the $GABA_B$ receptor to generate responses by the different $G_{i/o}$ subtypes in transfected HEK293 cells and in CGNs, then revealing the importance of studying GPCRs in their native environment.

## Lower agonist potencies for the endogenous $GABA_B$ receptor in neurons

We have further validated these G protein sensors by measuring the potency of different orthosteric $GABA_B$ receptor ligands (Fig. 3a) in both CGNs and HEK293 cells. Similar potencies were obtained for the $G_{i1}$ and $G_{oA}$ for baclofen and other agonists, and for the antagonist CGP64213 in CGNs (Fig. 3b–f, Supplementary Table 1). The $GABA_B$ receptor agonists APPA and SKF 97541 were more potent than GABA and baclofen in both CGNs (Fig. 3c–f) and HEK293 cells (Supplementary Fig. 4a), which was consistent with previous findings using the GTPγS assay in brain tissues[21]. $pEC_{50}$s obtained in CGNs were correlated with those measured in HEK293 cells (Fig. 3e, f), even though they were more than 10 times higher in HEK293 cells, except for baclofen where the difference is lower (Supplementary Table 1, Fig. 3e, f). Similar results were obtained in the hippocampal neurons and cortical neurons (Supplementary Fig. 4b–d and Supplementary Table 1). These differences in baclofen potencies observed between transfected HEK293 cells and native neurons were not due to the difference in receptor expression. Indeed, when the expression level $GABA_B$ receptors was similar to that in CGNs (Fig. 2b), the baclofen $pEC_{50}$ was similar to that observed with higher expression levels, and still significantly different from that measured in CGNs (Fig. 3g).

We have also validated our sensors for the analysis of positive allosteric modulators (PAMs). PAMs can synergistically enhance the activity of native neurotransmitter GPCRs in neurons[43] and have a greater potential for drug development with less side effect[44], including for the $GABA_B$ receptor[45]. It was recently demonstrated that these PAMs bind at the interface of the transmembrane domains of this dimeric receptor[46–48] (Fig. 3a). Here we showed that Rac BHFF increased strongly the potency of baclofen-induced $G_{i1}$ (Fig. 3h) and $G_{oA}$ protein (Supplementary Fig. 5a) response in CGNs. However, in contrast to what is observed in HEK293 cells[48,49] (Fig. 3h and Supplementary Fig. 5a–b), Rac BHFF alone (10 µM) did not increase the net BRET signal in CGNs (Fig. 3h and Supplementary Fig. 5a). It is nicely illustrated by concentration responses curves of the Rac BHFF obtained with transfected HEK293 cells but not so efficiently with neurons (Supplementary Fig. 5c). It suggests that the Rac BHFF agonist activity observed with the recombinant $GABA_B$ receptor is much higher than that measured for the receptor in its native environment. This slight agonist effect of Rac BHFF in CGNs is probably not due to the lower $GABA_B$ receptor expression in neurons compared to HEK293 cells. Indeed, the agonist activity of Rac BHFF can still be observed in HEK293 cells that expressed $GABA_B$ receptor at a similar level to that found in CGNs (Figs. 2b, 3h, Supplementary Fig. 5a).

Finally, we took advantage of our sensors to measure the constitutive activity[50], in the absence of agonist, of the $GABA_B$ receptor that is usually observed in recombinant systems[47,49,51–55]. This constitutive activity was reversed by the competitive antagonists known for its

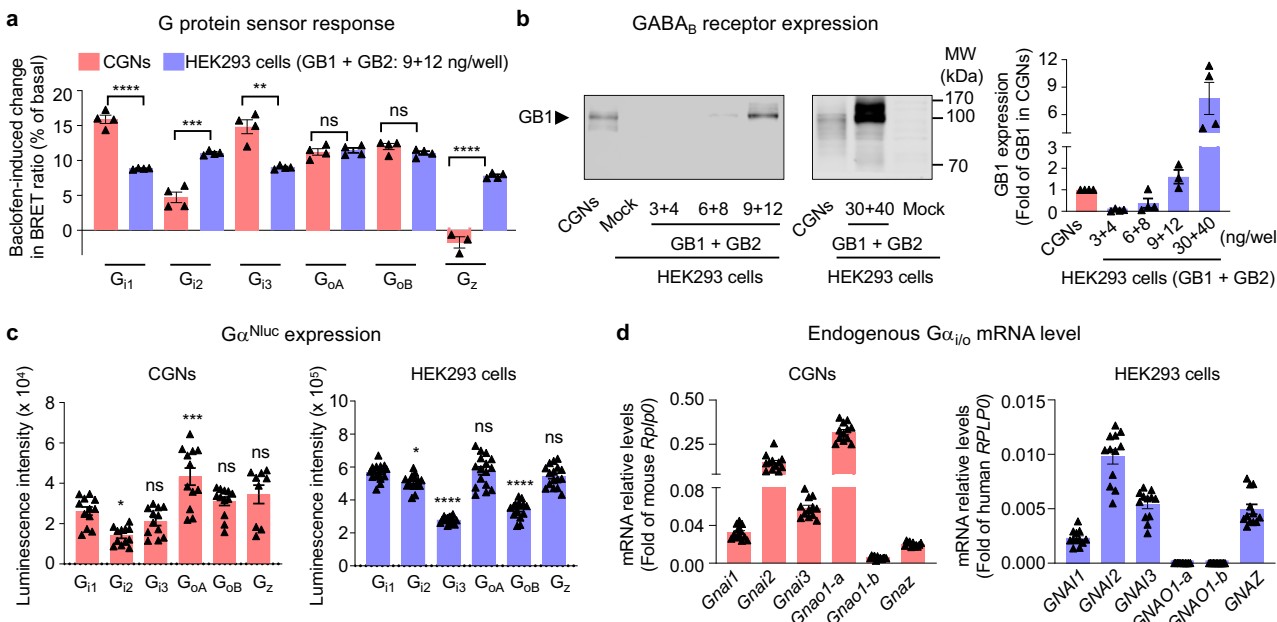

**Fig. 2 | Activity of endogenous GABA_B receptor detected by the G_{i/o} protein sensors. a** Baclofen-induced change in BRET ratio between $G\alpha^{Nluc}$ and $^{Venus}G\gamma_2$ in CGNs or transfected HEK293 cells for the indicated $G_{i/o}$ proteins, and expressed as percentage of the basal signal (($BRET_{basal}$-$BRET_{agonist}$/$BRET_{basal}$) x 100). CGNs were co-transfected with $^{Venus}G\gamma_2$ (50 ng) and the Nluc-tagged $G\alpha_{i1}$ (25 ng), $G\alpha_{i2}$ (75 ng), $G\alpha_{i3}$ (75 ng), $G\alpha_{oA}$ (25 ng), $G\alpha_{oB}$ (25 ng) or $G\alpha_z$ (25 ng), while HEK293 cells were co-transfected with GB1 (9 ng), GB2 (12 ng), $G\beta_1$ (10 ng), $^{Venus}G\gamma_2$ (10 ng) and the Nluc-tagged $G\alpha_{i1}$ (1 ng), $G\alpha_{i2}$ (3 ng), $G\alpha_{i3}$ (3 ng), $G\alpha_{oA}$ (1 ng), $G\alpha_{oB}$ (1 ng) or $G\alpha_z$ (1.5 ng) per well in 96-well plate. **b** Expression of GB1 subunit of GABA_B receptor in the cell membrane of CGNs, mock HEK293 cells, and HEK293 cells transfected with indicated amount of GB1 and GB2, detected by western blotting. Values are mean ± SEM normalized as fold of GB1 expression in CGNs from four independent experiments. **c** Expression of the Nluc-tagged $G\alpha$ transfected in CGNs or in HEK293 cells in (**a**), measured by the luminescent signal at 480 nm. The values of individual well are

shown. Values are mean ± SEM from four biologically independent experiments each performed in triplicates or quadruplicate in (**a**) and (**c**). Data in (**a**) are analysed using unpaired *t*-test (two-tailed) to determine significance. Data in (**c**) are analysed using one-way analysis of variance (ANOVA) with a Dunnett's *post-hoc* multiple comparison test to determine significance (compared with $G_{i1}$ group).
****$p < 0.0001$, ***$p < 0.001$, **$p < 0.01$ and *$p < 0.05$ and not significant (ns).
**d** Expression of the genes encoding indicate**d** $G\alpha$ in CGNs (mouse species: *Gnai1, Gnai2, Gnai3, Gnao-1, Gnao-2, Gnaz*) and HEK293 cells (human species: *GNAI1, GNAI2, GNAI3, GNAO-1, GNAO-2, GNAZ*). Values were determined by qRT-PCR and mean ± SEM normalized to *RplpO* (mouse) or *RPLPO* (humans) from three biologically independent experiments each performed in quadruplicate. The values of individual sample are shown. The raw data and *p*-values are available in source data provided as a Source Data file.

inverse agonist activity[49,51,55,56]. Accordingly, the competitive antagonist CGP54626 increased the basal BRET signal between $G\alpha$ and $G\beta\gamma$ for both the $G_{i1}$ and $G_{oA}$ sensors in the HEK293 cells (Supplementary Fig. 5d), as previously reported[49]. It is consistent with the properties of this inverse agonist to favour the $G\alpha\beta\gamma$ heterotrimer by stabilizing the inactive conformation of the receptor. Interestingly, this increase of BRET signal was not observed in the CGNs (Supplementary Fig. 5d), indicating the native GABA_B receptor has no or a very low constitutive activity that cannot be observed with our sensor. It is consistent with the weak intrinsic agonist activity of Rac BHFF in neurons.

Altogether, our results show that GABA_B receptor ligands have different potencies and efficacies in the cell lines compared to neurons, which cannot be explained by a difference in GABA_B receptor expression levels between the two cell types.

**Various efficacies of endogenous $G_{i/o}$-coupled GPCRs in neurons**
We then examined the activities of other endogenous $G_{i/o}$-coupled GPCRs in the CGNs. First, the agonist CP 55940 that actives both the cannabinoid receptor type 1 (CB1) and 2 (CB2) induced a strong change in both $G_{i1}$ and $G_{oA}$ BRET signal as observed for the GABA_B receptor (Fig. 4a). It was expected that it is due to the activation of CB1 since this receptor is highly expressed in cerebellum[33] compared to CB2 that is mainly expressed at the periphery[57]. When testing other GPCR agonists, a strong or significant change of BRET signal was measured with different ligands used at saturation concentrations for their expected receptors: brimonidine (also named UK-14304), pramipexole, ele-triptan and carbachol known to activate $\alpha_2$ adrenoceptor[58], D_{2/4}

receptor[59], 5-hydroxytryptamine receptor 1B/1D[60] and the M_{2/4} mus-carinic receptors[61], respectively. In contrast, agonists of the platelet-activating factor (PAF) receptor and δ/μ opioid receptor DAMGO failed to induce a BRET signal change (Fig. 4a). Dose-response of brimoni-dine, CP 55940, pramipexole and carbachol further confirmed the specific activation of these endogenous $G_{i/o}$-coupled receptors (Fig. 4b). Altogether, our data correspond to the receptors for which higher amount of mRNA were detected in these cultured CGNs at the developmental stage investigated[62].

Next, using $\alpha_2$ adrenoceptor we show that our G protein sensors can determine ligand efficacies, by evaluating the full agonist brimo-nidine and several reported partial agonists (clonidine, oxymetazoline, tizanidine and xylazine)[63]. In contrast, brimonidine exhibited the highest response of $G_{i1}$ and $G_{oA}$ protein in CGNs, while others showed less response (Fig. 5a). However, the discrimination between brimo-nidine and other agonists was not significantly observed in HEK293 cells transfected with $\alpha_{2A}$ adrenoceptor ($\alpha_{2A}AR$) (Fig. 5b). It was not due to the high level of expression of $\alpha_{2A}AR$, since no discrimination between brimonidine and other agonists was observed in HEK293 cells that expressed $\alpha_{2A}AR$ at a similar level to that found in CGNs (Fig. 5c). In these conditions (15 ng of $\alpha_{2A}AR$ cDNA/well), the brimonidine $pEC_{50}$ was similar to that observed with higher expression levels (Supple-mentary Fig. 6a), and still significantly different from that measured in CGNs (Fig. 5d), as observed previously for the GABA_B receptor (Fig. 3g). Finally, the data showed a significant difference in the G protein responses mediated by the different $G_{i/o}$ subtypes triggered by the recombinant $\alpha_{2A}AR$ in HEK293 cells and the endogenous one in

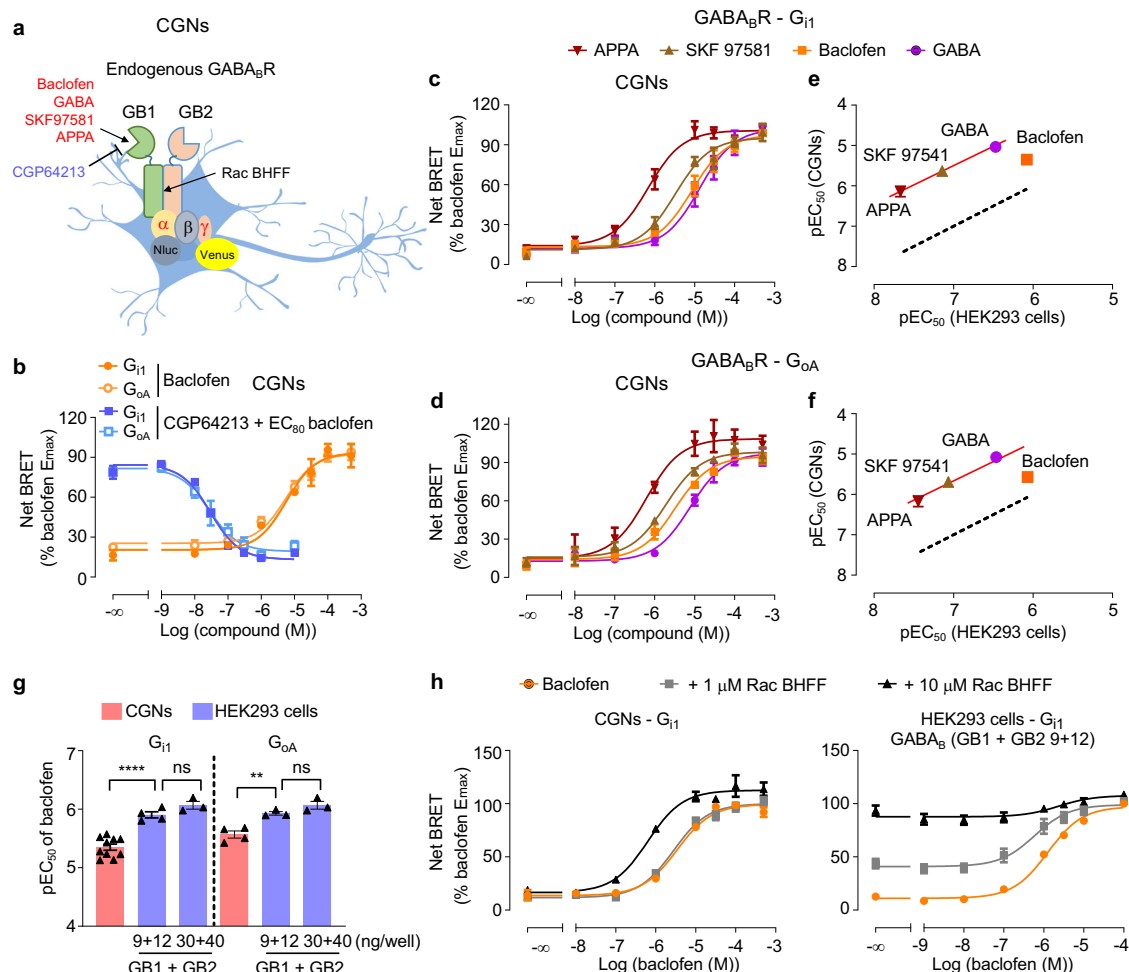

**Fig. 3 | Lower agonist potencies of the endogenous GABA$_B$ receptor in CGNs.**
**a** Scheme of the GABA$_B$ receptor, made of GB1 and GB2 subunits, in CGNs and its various specific ligands (agonist, red; antagonists, blue; PAM, black). **b** Change of BRET signal induced by the GABA$_B$ receptor antagonist CGP64213 in presence of 20 µM baclofen (EC$_{80}$) or different doses of baclofen in CGNs measured by G$_{i1}$ or G$_{oA}$ sensor. **c**, **d** Change of BRET ratio induced by various GABA$_B$ receptor agonists measured by G$_{i1}$ (**c**) or G$_{oA}$ (**d**) sensor. **e**, **f** Correlation of the agonist potencies (pEC$_{50}$) between HEK293 cells and CGNs determined by the G$_{i1}$ (**e**) or G$_{oA}$ (**f**) BRET sensors. Dotted line is the correlation of pEC$_{50}$ determined with HEK293 cells. Red line is fit of pEC$_{50}$ in CGNs with the same slope as the dotted line. **g** The pEC$_{50}$ of baclofen in CGNs and transfected HEK293 cells with indicated amount of GB1 and GB2 measured by G$_{i1}$ or G$_{oA}$ sensor. Data are mean ± SEM from at least three

biologically independent experiments each performed in triplicate and analysed using one-way ANOVA with a Dunnett's *post-hoc* multiple comparison test to determine significance. The n number of G$_{i1}$ and G$_{oA}$ group for CGNs, and HEK293 cells transfected with the indicated amount of GB1 and GB2 are 10, 4, 3, and 4, 3, 3, respectively. ****$p < 0.0001$, **$p < 0.01$ and not significant (ns). **h** Change of BRET ratio induced by baclofen with or without indicated concentrations of the GABA$_B$ PAM Rac BHFF in transfected HEK293 cells and CGNs measured by G$_{i1}$ sensor. Data are normalized to maximum of baclofen response and mean ± SEM from at least three biologically independent experiments each performed in triplicate in (**b**–**d**) and (**h**). **b** $n = 3$; **c** $n = 6$; **d** $n = 4$; **h** CGNs $n = 3$, HEK293 cells $n = 4$. The raw data and $p$-values are available in source data provided as a Source Data file.

CGNs (Fig. 5e and Supplementary Fig. 6b). Of note, similarly to experiments above with GABA$_B$ receptor (Fig. 2c), the expression of the different Gα$^{Nluc}$ proteins was within a factor of three in each type of cell (Supplementary Fig. 6c).

Our results demonstrated that G$_{i/o}$ protein Nluc-biosensors can be widely applied to different endogenous G$_{i/o}$-coupled GPCRs and evaluate the ligand efficacies and G protein coupling profile in neurons. They also further revealed the importance of studying a GPCR in its native environment.

### Difference between G$_i$ and G$_o$ proteins revealed for the endogenous CB1 receptor
Finally, we took advantage of the major change in BRET signal observed for the cannabinoid receptor to investigate the G protein response in CGNs. While level of expression of the CB1 receptor in the two systems was similar in western blotting experiments (Fig. 6a), a higher CB1 receptor-mediated G protein response was observed with

G$_{i1}$ and G$_{i3}$ only, with the native receptor in CGNs compared to the recombinant one in HEK293 cells (Fig. 6b, Supplementary Fig. 7a). Similar responses were observed with the other G$_{i/o}$ subtype sensors between two types of cells. Again, as with the GABA$_B$ receptor (Fig. 2b) and α$_{2A}$AR (Supplementary Fig. 6c), the expression of the different Gα$^{Nluc}$ proteins was within a factor of three in each type of cell (Supplementary Fig. 7b). And a small overexpression of the CB1 receptor by transfection of CGNs with the recombinant CB1 receptor gave similar results (Supplementary Fig. 7c, d). Interestingly, our G protein sensors revealed a higher potency of CP 55940, Win 55,212-2 and Bay 59-3074 in CGNs compared with transfected HEK293 cells for both G$_{i1}$ and G$_{oA}$ response (Fig. 6c, d and Supplementary Table 2). It is different from GABA$_B$ and α$_2$AR, which showed higher agonist potencies in transfected HEK293 cells. Of note, when taking Win 55,212-2 as a reference ligand, Bay 59-3074 showed G$_{oA}$ bias in both CGNs and HEK293 cells with a bias factor of 1.6 and 1.3 respectively. CP 55940 showed a G$_{i1}$ bias in CGNs and a G$_{oA}$ bias in HEK293 cells with a bias factor of 1.6 and 1.4

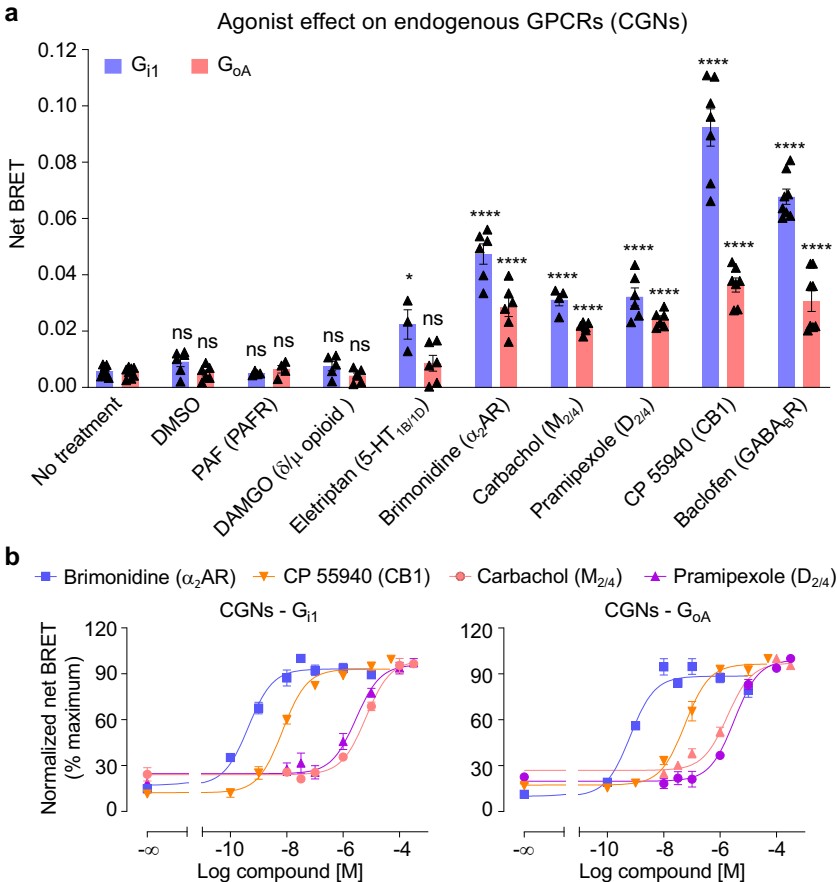

**Fig. 4 | $G_{i/o}$ protein sensors report the activity of other endogenous $G_{i/o}$-coupled GPCRs in CGNs. a** Net BRET signal of the $G_{i1}$ and $G_{oA}$ sensors in CGNs induced by specific agonists of various $G_{i/o}$-coupled GPCRs using a saturating concentration of the indicated ligands (PAF, 10 μM; DAMGO, 100 μM; eletriptan, 100 μM; brimonidine, 10 μM; carbachol, 100 μM; pramipexole, 100 μM; CP 55940, 20 μM and baclofen, 100 μM). Values are mean ± SEM from at least biologically independent experiments each performed in triplicate. The *n* number of the indicated treatments in $G_{i1}$ and $G_{oA}$ are 6, 4, 5, 4, 6, 4, 6, 7, 8, and 8, 6, 4, 6, 6, 6, 6, 7, 8, respectively. Data are analysed using one-way ANOVA with a Dunnett's *post-hoc* multiple comparison test to determine significance (compared with no treated

group). ****$p < 0.0001$, *$p < 0.05$ and not significant (ns). **b** Dose-response of the agonists brimonidine, CP 55940, carbachol and pramipexole for the $\alpha_2$ adrenoceptor, CB1, muscarinic $M_{2/4}$ and dopamine $D_{2/4}$ receptors, respectively, in CGNs co-transfected with $G\alpha_{i1}^{Nluc}$ and $^{Venus}G\gamma_2$ or $G\alpha_{oA}^{Nluc}$ and $^{Venus}G\gamma_2$. Data are normalized to maximum response of each compound and mean ± SEM from at least three biologically independent experiments each performed in triplicate. $G_{i1}$, brimonidine ($n = 3$), CP 55940 ($n = 4$), carbachol ($n = 4$) and pramipexole ($n = 4$); $G_{oA}$, brimonidine ($n = 3$), CP 55940 ($n = 3$), carbachol ($n = 4$) and pramipexole ($n = 3$). The raw data and *p*-values are available in source data provided as a Source Data file.

respectively, indicating a difference between $G_{i1}$ and $G_{oA}$ proteins revealed by CP 55940 (Fig. 6c, d and Supplementary Tables 2 and 3).

## Major influence of the Gγ subunit on the G protein coupling in CGNs

The influence of Gβγ subunit composition on G protein coupling[64,65] was largely reported even though the underlying molecular mechanism is largely unknown. Therefore, we have compared sensors composed by $G\alpha^{Nluc}$ and different Gγ including $^{Venus}G\gamma_2$, $^{Venus}G\gamma_8$ or $^{Venus}G\gamma_9$, which are well-used in the previously reported G protein BRET sensors[25,36]. $^{Venus}G\gamma_8$ or $^{Venus}G\gamma_9$ belong to two different groups of Gγ subunits (Supplementary Fig. 8). $G\gamma_8$, like $G\gamma_2$, is well expressed in some regions of the brain, in contrast to $G\gamma_9$ that seems restricted to retina[66]. Interestingly, a major difference in G protein response profile of the GABA$_B$ receptor and $\alpha_2$ adrenoceptor was measured depending on the Gγ subunits used (Fig. 7a–c and Supplementary Fig. 9a–d), that clearly shows the impact of the Gγ subunit in these GPCR-mediated responses.

## Discussion

Our study compares the pharmacological and G protein responses of native, versus recombinant $G_{i/o}$-coupled receptors in primary neurons

and in heterologous cells, respectively. Such an analysis was made possible thanks to the sensitive BRET-based $G_{i/o}$ protein sensors. These sensors allow a simple analysis of different types of GPCR ligands on each $G_{i/o}$ protein subtypes in primary neurons in a medium throughput format (in 96-well plate). For the same G protein sensor, our study reveals important differences in agonist potencies between neurons and heterologous HEK293 cells (Fig. 8a). GABA$_B$ and $\alpha_2$ adrenergic receptor agonists displayed higher potencies in cell lines versus neurons even at similar receptor expression levels, in contrast to CB1 agonist. And when comparing the profile of response of different G protein sensors in the same cellular system, major differences of potencies can be observed, such as for a CB1 agonist (CP 55940) that shows high potency in mediating $G_{i1}$ response compared to $G_{oA}$ in neurons. These data highlight the importance of evaluating the activities of endogenous GPCRs in their native environment, and of developing biosensors allowing such analyses.

In order to be able to compare the properties of recombinant and native $G_{i/o}$-coupled GPCRs, we needed sensors for the various $G_{i/o}$ proteins compatible with measurements of their activity in primary neurons. Here we showed that Nluc-based BRET sensors, could do so, most likely due to the high emission of Nluc, compared to Rluc8 used previously[24,25], since the efficacy of transfection of these constructs in

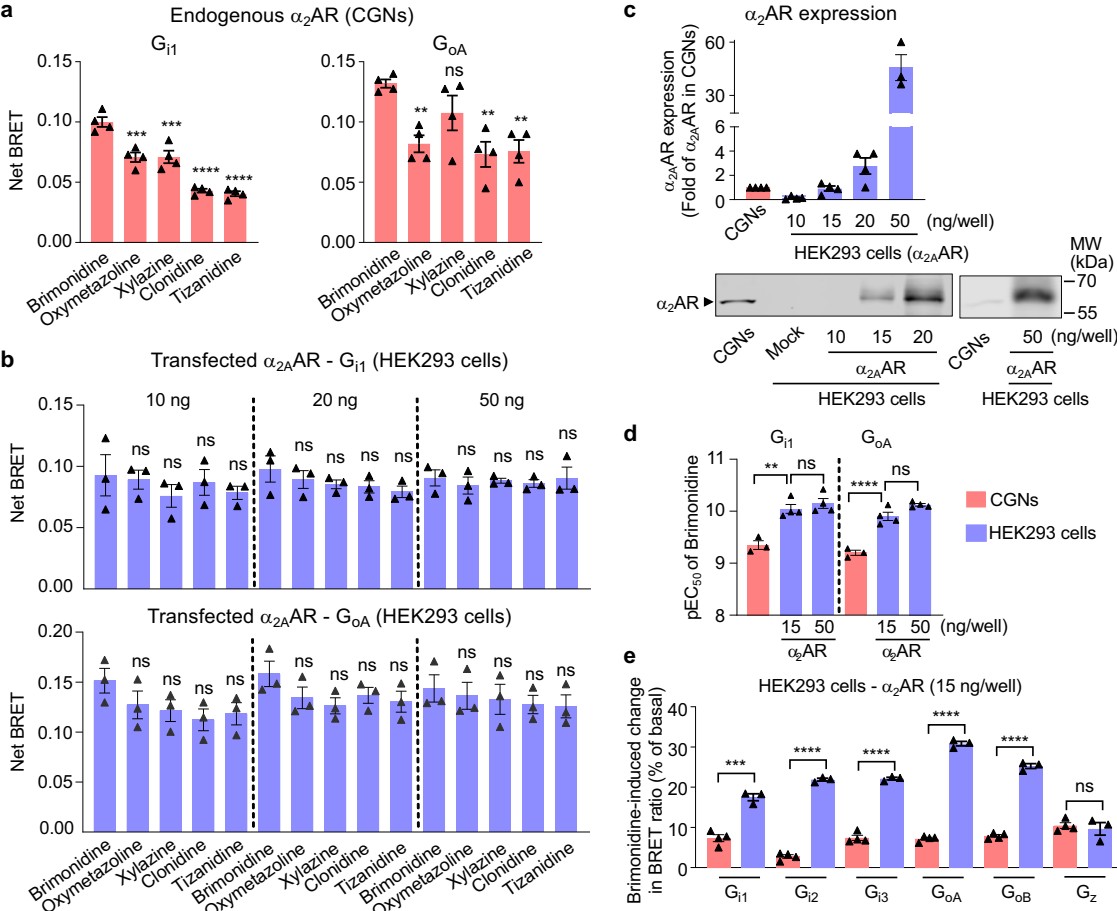

**Fig. 5 | Activity of endogenous $\alpha_2$AR detected by the $G_{i/o}$ protein sensors.**
**a** Effect of the indicated $\alpha_2$AR agonists on the net BRET of the $G_{i1}$ and $G_{oA}$ sensors in CGNs co-transfected with $G\alpha_{i1}^{Nluc}$ and $^{Venus}G\gamma_2$ or $G\alpha_{oA}^{Nluc}$ and $^{Venus}G\gamma_2$ (amounts of cDNA as in Fig. 2a). **b** Effect of the indicated $\alpha_2$AR agonists on the net BRET of the $G_{i1}$ and $G_{oA}$ sensors in HEK293 cells co-transfected with the mouse $\alpha_{2A}$AR (10 ng, 20 ng or 50 ng), and $G\beta_1$, $^{Venus}G\gamma_2$ and $G\alpha_{i1}^{Nluc}$ or $G\alpha_{oA}^{Nluc}$ as in Fig. 2a. Saturating concentrations of brimonidine (10 µM), oxymetazoline (100 µM), xylazine (50 µM), clonidine (100 µM), tizanidine (100 µM) were used in (**a**, **b**). **c** Detection of $\alpha_2$AR in cell membranes of CGNs, mock-transfected HEK293 cells and HEK293 cells transfected with indicated amount of $\alpha_2$AR, by western blotting. Values are mean ± SEM normalized as fold of $\alpha_2$AR expression in CGNs from four independent experiments. **d** The $pEC_{50}$ of brimonidine in CGNs ($n = 3$) and transfected HEK293 cells

($n = 4$) with indicated amount of $\alpha_2$AR measured by $G_{i1}$ or $G_{oA}$ sensor. **e** Percentage of change in BRET ratio between $G\alpha^{Nluc}$ and $^{Venus}G\gamma_2$ induced by brimonidine between $G\alpha^{Nluc}$ and $^{Venus}G\gamma_2$ in HEK293 cells ($n = 3$) ($\alpha_2$AR: 15 ng/well per 96-well plate) or CGNs ($n = 4$) for the indicated $G\alpha_{i1}$, $G\alpha_{i2}$, $G\alpha_{i3}$, $G\alpha_{oA}$, $G\alpha_{oB}$ or $G\alpha_z$ sensors (amounts of cDNA as in Fig. 2a). Values are mean ± SEM from at least three biologically independent experiments each performed in triplicate or quadruplicate in (**a**, **b**, **d**, **e**). **a**, $n = 4$; **b**, $n = 3$; (**d**, **e**), CGNs, $n = 3$; HEK293 cells, $n = 4$. Data are analysed using one-way ANOVA with a Dunnett's *post-hoc* multiple comparison test to determine significance (compared with brimonidine in **a**, **b**). Data are analysed using unpaired *t*-test (two-tailed) in (**e**). ****$p < 0.0001$, ***$p < 0.001$, **$p < 0.01$ and not significant (ns). The raw data and *p*-values are available in source data provided as a Source Data file.

primary neurons remains very low (around 2%). The principle of our optical biosensors is not novel, since the rearrangement between G$\alpha$ and G$\beta\gamma$ upon receptor activation using resonance energy transfer techniques was previously reported by FRET[23], BRET[24,25] including with the use of Nluc[36]. However, these biosensors have never been tested in primary neurons. Interestingly, other kind of biosensors compatible with the detection of endogenous G protein activity were recently reported[22,36]. Among them, the unimolecular "BERKY" sensors were used in live primary cells but they required the use of a viral vector. In addition, these sensors were not able to discriminate between different G protein subtypes, such as between $G_{i1}$, $G_{i2}$, $G_{i3}$, $G_{oA}$, $G_{oB}$ and $G_z$ since it is a unimolecular sensor using a peptide recognizing the GTP form of these different G proteins. Accordingly, they cannot be used to analyse the influence of the different G$\beta\gamma$ subunits as our sensors and the previous BRET sensors[24,25,36] do. Instead, our sensors nicely revealed specific responses of each $G_{i/o}$ protein triggered by some native GPCRs in primary neurons, which are different from what can be observed in heterologous cells. They work well for the most abundant

neuronal GPCRs, but would likely need to be further improved for receptors expressed at lower levels.

We have to be cautious when comparing the profile of response of different G protein sensors in the same cellular system (Figs. 7 and 8b). Indeed, each G protein BRET sensor might have its own properties, with its specific range of signal, even though they were constructed similarly, and $G\alpha_{i/o}$ subtypes have a high sequence identity[37]. G protein sensor response can result either from a complete dissociation between G$\alpha$ and G$\beta\gamma$ but also from a repositioning of G$\beta\gamma$ relative to G$\alpha$ without a real dissociation[23]. Within the $G_{i/o}$ protein family, using similar FRET-based G protein sensors between G$\alpha$ and G$\beta\gamma$, Frank et al. have proposed that receptor-activated $G_{i1}$, $G_{i2}$, $G_{i3}$ and $G_z$ undergo subunit rearrangement rather than subunit dissociation, whereas $G_o$ proteins either rearrange with a very distinct pattern or dissociate during activation[67]. But even though there is good evidence that the activated G$\alpha$ and G$\beta\gamma$ subunits actually loose affinity to each other and exchange faster depending of the G proteins, there is no evidence that the subunits separate completely in intact cells[68].

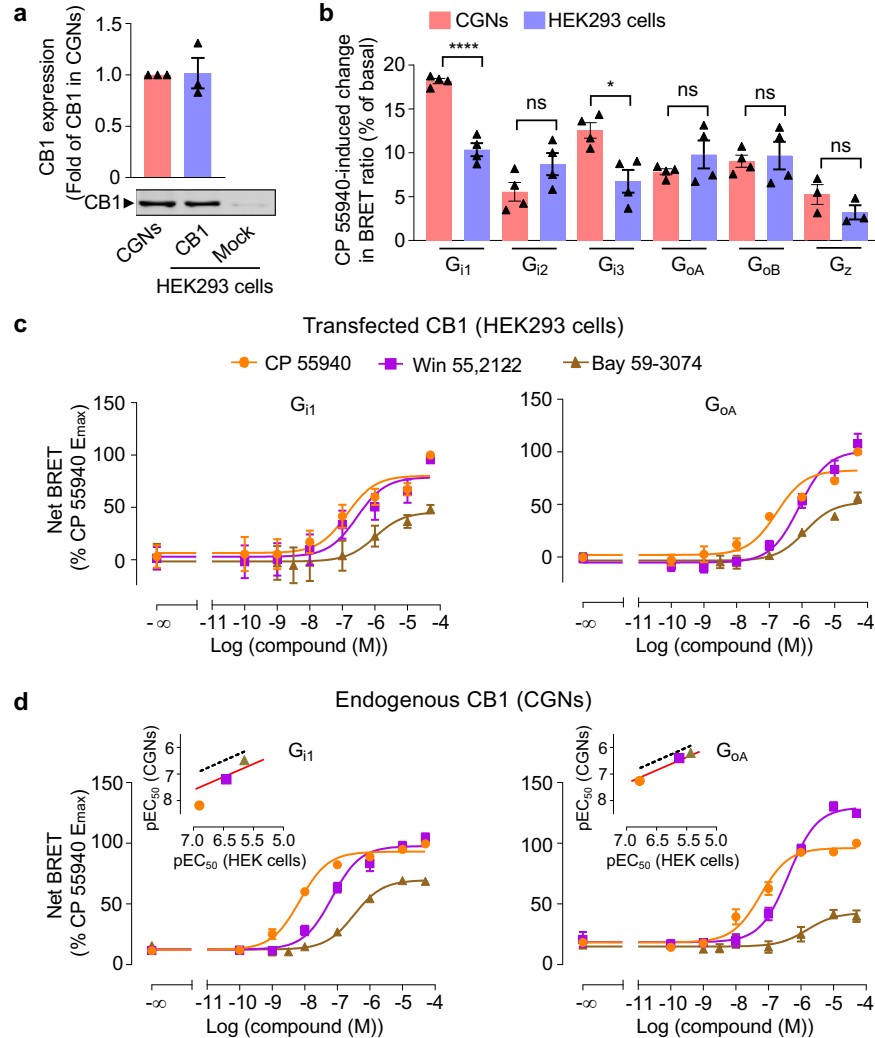

**Fig. 6 | Difference of $G_{i1}$ and $G_{oA}$ responses by cannabinoid receptor CB1 agonists. a** Detection of CB1 receptor in cell membranes of CGNs, mock-transfected HEK293 cells and HEK293 cells transfected with mouse CB1 cDNA (80 ng/well in 96-well plate), by western blotting. Values are mean ± SEM normalized as fold of CB1 expression in CGNs from three independent experiments. **b** Percentage of change in BRET ratio between $G\alpha^{Nluc}$ and $^{Venus}G\gamma_2$ induced by CP 55940 in HEK293 cells (CB1 cDNA; 80 ng/well per 96-well plate) or CGNs for the indicated $G\alpha_{i1}$, $G\alpha_{i2}$, $G\alpha_{i3}$, $G\alpha_{oA}$, $G\alpha_{oB}$ or $G\alpha_z$ sensors (amounts of cDNA as in Fig. 2a). Values are mean ± SEM from four biologically independent experiments each performed in triplicate. Data are analysed using unpaired *t*-test (two-tailed) to determine significance. ****$p < 0.0001$, *$p < 0.05$ and not significant (ns). **c, d** Change of BRET signal between $G\alpha^{Nluc}$ and $^{Venus}G\gamma_2$ induced by the indicated CB1 receptor agonists in HEK293 cells and CGNs measured by $G_{i1}$ and $G_{oA}$ sensors. The transfection was the same as in (**b**). *Inset*, correlation of the agonist potencies (pEC$_{50}$) between HEK293 cells and CGNs determined by the G protein sensors. Dotted lines are the correlation of pEC$_{50}$ determined with HEK293 cells. Red lines are the fit of pEC$_{50}$ in CGNs with the same slope as the dotted lines. Data are normalized to maximum CP 55940 response and are mean ± SEM from at least three biologically independent experiments each performed in triplicate (HEK293 cells: $G_{i1}$, CP 55940 ($n = 5$), Win 55,212-2 ($n = 5$) and Bay 59-3074 ($n = 4$); $G_{oA}$, CP 55940 ($n = 5$), Win 55,212-2 ($n = 4$) and Bay 59-3074 ($n = 4$). CGNs: $G_{i1}$, CP 55940 ($n = 4$), Win 55,212-2 ($n = 4$) and Bay 59-3074 ($n = 4$); $G_{oA}$, CP 55940 ($n = 4$), Win 55,212-2 ($n = 4$) and Bay 59-3074 ($n = 3$)). The raw data and *p*-values are available in source data provided as a Source Data file.

When analysing the same G protein sensor, our study reveals major differences in G protein responses and agonist potencies between neurons and heterologous HEK293 cells. The difference in the profiles of G protein response using different G$\gamma$ subunits (G$\gamma_2$, G$\gamma_8$ or G$\gamma_9$) (Fig. 7) shows clearly the impact of G$\gamma$ in CGNs. It shows that not only the G$\alpha$ subunit is important, but also the G$\beta$ and G$\gamma$, consistent with other recent studies[65,66]. Therefore, it brings to our attention that the G protein-mediated response also depends detected on the bio-sensor components used, which is the limitation of biosensors that only detect the response of a well-defined G protein. Meanwhile, the difference can be due to the intrinsic properties of the system under study. For example, the association of these receptors with specific protein partners that may influence their coupling to some G$\alpha_i$ subtypes, present in neurons but not in HEK293 cells. Indeed, specific proteins interacting with the GABA$_B$ receptor were identified in brain tissues[15,69], while they are not expressed in HEK293 cells. For example, the soluble form of APP, AJAP-1 and PIANP bind to the extracellular sushi domain of GABA$_B$[70,71], TRPV1 channels interact in the membrane[72], and the intracellular KCTDs and 14-3-3 proteins[73] modulate GABA$_B$ receptor downstream signalling. The KCTD proteins are constitutively associated with GABA$_B$ and they control its kinetics of activation[73].

Our sensors showed a difference between Gi and Go proteins revealed by a CB1 agonist CP 55940, which has a higher potency in generating $G_{i1}$ than $G_{oA}$ responses in neurons, relative to the other agonists tested. Indeed, such biased signalling between Gi and Go proteins have already been reported upon agonist stimulation[12,13,74], by allosteric modulators[49] or as a result of genetic variation[75]. In our study, the molecular bases of this difference in agonist effect is unclear. CP 55940 and Win 55212-2 ($K_i$ 1.1 nM and 62.3 nM for CB1 receptor,

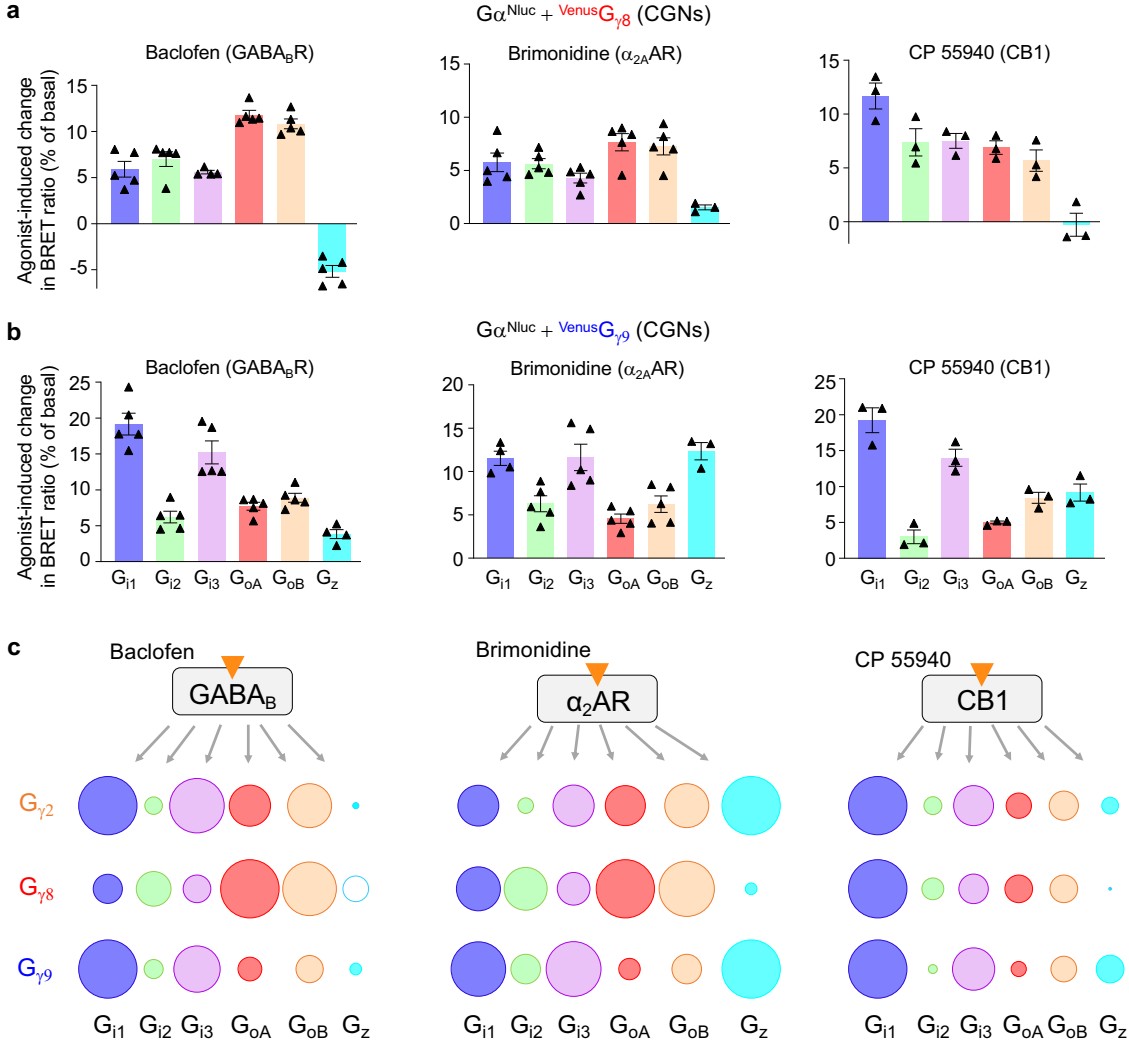

**Fig. 7 | Gγ subunit influences G protein responses in CGNs. a, b** Percentage of change in BRET ratio between Gα$^{Nluc}$ and $^{Venus}$Gγ$_8$ (**a**) or $^{Venus}$Gγ$_9$ (**b**) induced by baclofen, brimonidine and CP 55940 in CGNs for the indicated G$_{i/o}$ proteins. CGNs were co-transfected with the indicated $^{Venus}$Gγ and Nluc-tagged Gα$_{i1}$, Gα$_{i2}$, Gα$_{i3}$, Gα$_{oA}$, Gα$_{oB}$ or Gα$_z$. Values are mean ± SEM from biologically independent experiments (baclofen and brimonidine, $n = 5$; CP 55940, $n = 3$) each performed in triplicate. **c** Scheme illustrating the difference in G$_{i/o}$ (G$_{i1}$, G$_{i2}$, G$_{i3}$, G$_{oA}$, G$_{oB}$ and Gα$_z$) and indicated Gγ protein (Gγ$_2$, Gγ$_8$ and Gγ$_9$) responses in CGNs for the receptors GABA$_B$, CB1 and α$_2$AR in presence of the indicated agonists. For each receptor and

G$_{i/o}$ sensor, the size of the circle is the percentage of change in BRET ratio upon the agonist stimulation, for GABA$_B$ (Figs. 2a and **a**, **b**), α$_2$AR (Figs. 5e and **a**, **b**) and CB1 (Figs. 6b and **a**, **b**) normalized to the G$_{i/o}$ sensor that has the highest response: for Gγ$_2$, G$_{i1}$ for both the GABA$_B$ and CB1, and Gα$_z$ for α$_2$AR; for Gγ$_8$, G$_{oA}$ for GABA$_B$ and α$_2$AR and G$_{i1}$ for CB1; and for Gγ$_9$, G$_{i1}$ for GABA$_B$ and CB1, and Gα$_z$ for α$_2$AR. For Gγ8, the empty circle indicates an increase in the BRET ratio in contrast to other conditions where a decrease BRET ratio is measured. The raw data and $p$-values are available in source data provided as a Source Data file.

respectively)[76,77] have different scaffolds. Their differential effects might be explained by the ligand-binding kinetics[78], by specific active conformation stabilized by the agonist[79], or due to specific components associated to the receptor in these neurons, better stabilizing the CP 55940/CB1/Gi complex.

Our sensors also revealed larger differences in some agonist potencies in neurons compared to transfected HEK293 cells. This is nicely illustrated for the CB1 and GABA$_B$ agonists CP 55940 and APPA, respectively, that show a higher potency than the other agonists in neurons (Fig. 8a). In addition and unexpectedly, no intrinsic agonist activity of the PAM Rac BHFF was detected with the native GABA$_B$ receptor in contrast to the recombinant one in HEK293 cells. Finally, the important differences in GABA$_B$ receptor agonist potencies that are higher in transfected HEK293 cells compared to neurons are most probably not due to the higher expression in the transfected cells in our study. Indeed, a 10-times higher affinity for GABA in neurons compared to HEK cells[80] and similar agonist potencies in GTPγS

binding experiments in the two systems[81] were previously reported. Possible explanations might come from the environment of the receptor in neurons, such as its compartmentalization and its interaction with extracellular and intracellular proteins[15,50], including other endogenous receptors.

Our sensors also revealed specific ligand-independent G protein response in neurons compared to HEK293 cells. Indeed, no detectable constitutive activity of the GABA$_B$ receptor was observed in CGNs in contrast to transfected cell lines[49,51,52]. As discussed above, this lack of constitutive activity might be explained by the environment of the receptor in neurons. The importance of the intracellular protein partners for the constitutive activity of a GPCR was nicely illustrated for the postsynaptic metabotropic glutamate receptor mGlu5. Its constitutive activity was maintained low in neurons by the long isoform of Homer, while it is revealed when the short isoform of Homer is expressed[82].

In conclusion, our study reveals the importance of evaluating the pharmacological and G protein response properties of native GPCRs in

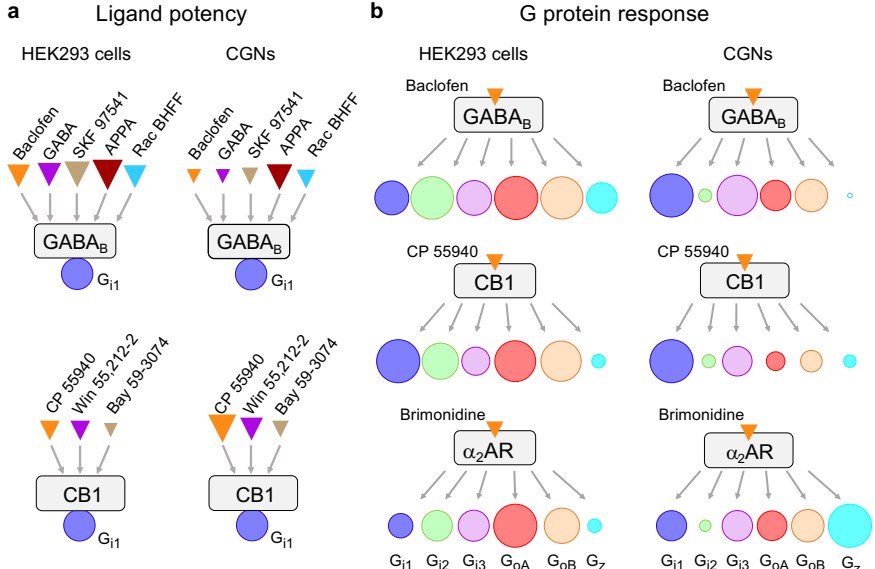

**Fig. 8 | Major differences in agonist potencies and the G protein response profile for some GPCRs between heterologous and native cells. a** Scheme illustrating the difference in agonist or PAM potencies for the $GABA_B$ and CB1 receptors between HEK293 cells and CGNs for the $G_{i1}$ sensor. For each cell and receptor, the relative size of the triangles is according to the $pEC_{50}$ values of the indicated agonists. **b** Scheme illustrating the difference in $G_{i/o}$ protein response ($G_{i1}$, $G_{i2}$, $G_{i3}$, $G_{oA}$, $G_{oB}$ and $G_z$) between CGNs and HEK293 cells for the receptors GABA$_B$, CB1 and $\alpha_2$AR in presence of the indicated agonists, when $G\gamma_2$ is used for all these $G_{i/o}$ sensors. For each cell and receptor, the size of the circle is the percentage of change in BRET ratio upon the agonist stimulation for GABA$_B$ (Fig. 2a), $\alpha_2$AR (Fig. 5e) and CB1 (Fig. 6b) normalized to the $G_{i/o}$ sensor that has the highest response: in HEK293 cells, $G_{oA}$ for GABA$_B$, $\alpha_2$AR and $G_{i1}$ for CB1, and in CGNs, $G_{i1}$ for both the GABA$_B$ and CB1, and $G_z$ for $\alpha_2$AR.

primary cells to better understand their signalling, identify and characterize ligands expected to have therapeutic effects. It also reveals the powerfulness of using G protein sensors compatible with native cells, though biosensors compatible with single cell analysis and imaging will certainly help revealing the potential cell-to-cell or sub-compartment heterogeneity in GPCR and G protein coupling[83,84]. In the future, methods measuring the activation of a specific endogenous G protein in live cells should be envisioned. Regardless of the criticism of our approach, our study clearly reveals that data generated in recombinant systems should be taken with caution before going further into pre-clinical and clinical development of drugs candidates only characterized this way.

## Methods

### Materials

GABA (Cat. A2129), CP 55940 (Cat. C1112) and carbachol (PHR1511) were purchased from Sigma-Aldrich (Shanghai, China). R-baclofen (Cat. 0796), SKF 97581 (Cat. 0379) and CGP54626 (Cat. 1088) were obtained from Tocris Biosciences (Shanghai, China). 3-APPA (Cat. ab120329) was purchased from Abcam (Shanghai, China). Win 55,212-2 (T4458), Bay 59-3074 (T3699), DAMGO (T4351), eletriptan HBr (T0216), pramipexole 2HCl monohydrate (T6951), brimonidine tartrate (T6422), tizanidine hydrochloride (T0290), oxymetazoline hydrochloride (T0252), xylazine (T7046), clonidine hydrochloride (T1247), and 1-oleoyl lysophosphatidic acid sodium (T21654) were purchased from TargetMol (Shanghai, China). PAF C-16 (74389-68-7) was purchased from Santa Cruz Biotechnology (Shanghai, China). CGP64213 was a gift from Prof. Fajun Nan (Shanghai Institute of Materials Medicine, China). Coelenterazine h (Cat. S2001) and furimazine (Cat. N1120) were purchased from Promega (Madison, WI, USA).

### Plasmids

The coding sequences of Nluc and Rluc8 were PCR-amplified and inserted into the coding sequence of human $G\alpha_{i1}$, $G\alpha_{i2}$, $G\alpha_{i3}$, $G\alpha_{oA}$ and $G\alpha_{oB}$ (between residues 91 and 92) and $G\alpha_z$ (between residues 113 and 114) in pcDNA3.1, using the flexible linkers SGGGGS and SGGGEF, before and after the sequence of Nluc, as shown in Supplementary Figs. 10c, d and 12. The pRK5 plasmid encoding the rat GABA$_B$ subunits (HA-tagged GABA$_{B1}$ (GB1) and Flag-tagged GABA$_{B2}$ (GB2)) and pcDNA3.1 encoding the human $G\beta_1$ and human $^{Venus}G\gamma_2$ were provided by the Institut de Génomique Fonctionnelle (Montpellier, France). The cDNA of mouse $\alpha_{2A}$AR, mouse CB1, human $G\gamma_8$ and human $G\gamma_9$ were bought from Miaoling Bio (Wuhan, China) and inserted in pcDNA3.1. Venus was inserted at the N-terminus of $G\gamma_2$, $G\gamma_8$ and $G\gamma_9$ between the HindIII and KpnI restriction enzymatic sites and the linker KLGT between Venus and the $G\gamma$ subunits was used. The schematic representation of the constructs (Supplementary Fig. 10) and their protein sequences (Supplementary Figs. 11–13) were shown.

### HEK293 cell culture and transfection

HEK293 cells (ATCC, CRL-1573, lot: 3449904) were cultured in DMEM supplemented with 10% FBS at 37 °C in a humidified incubator containing 5% $CO_2$. For transfections, cells were suspended and transfected using Lipofectamine 2000 with the appropriate expression constructs as previously described[49]. The cDNA amounts used per well in 96-well plate were as following unless indicated in figure legends: rat GABA$_B$ receptor, GB1 (3 ng, 6 ng, 9 ng, 30 ng) and GB2 (4 ng, 8 ng, 12 ng, 40 ng), mouse $\alpha_{2A}$AR (10 ng, 15 ng, 20 ng, 50 ng), mouse CB1 receptor (80 ng), $G\alpha_{i1}^{Nluc}$ (1 ng), $G\alpha_{i2}^{Nluc}$ (3 ng), $G\alpha_{i3}^{Nluc}$ (3 ng), $G\alpha_{oA}^{Nluc}$ (1 ng), $G\alpha_{oB}^{Nluc}$ (1 ng), $G\alpha_z^{Nluc}$ (1.5 ng), $G\beta_1$ (10 ng) and $^{Venus}G\gamma_2$ (10 ng). The ratio of DNA to Lipofectamine 2000 ratio was 1:2. After a 24 h culture in 96-well plates, the cells were ready for the bioluminescence resonance energy transfer (BRET) experiments.

### Primary neuron culture and transfection

For the primary culture of neurons, all experiments were specifically designed to minimize the number of animals used and were approved by the Animal Experimentation Ethics Committee of the College of Life Science and Technology, Huazhong University of Science and

Technology, Wuhan, China. Kunming mice were obtained from the Center for Disease Control and Prevention of Hubei Province. The mice were raised in a specific pathogen free (SPF) environment with an ambient temperature of 18–22 °C, a humidity of 50%-60%, and a 12 h light-dark cycle.

Primary cerebellar granule neuronal cultures were prepared from one-week-old newborn Kunming mice as previously described[39]. The dissected tissue was gently triturated after Versene (15040066; Gibco, Shanghai, China) treatment for 5 min at 37 °C, and the homogenate was centrifuged at 170 g for 5 min. In the meantime, mixtures of the DNA and Lipofectamine 2000 (Ref 11668019; Thermo Fisher Scientific, Shanghai, China) in a 1:3 ratio in an Opti-minimal essential medium (Opti-MEM) (Ref 31985070; Thermo Fisher Scientific, Shanghai, China) were prepared following the manufacturer's protocols and as previously described[42]. The pellet was re-suspended in culture medium DMEM-F12 (Ref 11320-033; Gibco) supplemented with 2 mM glutamine, 30 mM KCl, 100 U/mL penicillin, 100 µg/mL streptomycin, and 10% foetal bovine serum (FBS) and seeded into 96-well plates (100 µL/well) previously coated with poly-L-ornithine (Sigma-Aldrich). Then, the mixtures of DNA and Lipofectamine 2000 (50 µL/well) were added to the wells. BRET experiments were performed three or four days after transfection. BRET experiments were performed at DIV3 or DIV4. The cDNA amounts used per well in 96-well plate were as following: $G\alpha_{i1}^{Nluc}$ (25 ng), $G\alpha_{i2}^{Nluc}$ (75 ng), $G\alpha_{i3}^{Nluc}$ (75 ng), $G\alpha_{oA}^{Nluc}$ (25 ng), $G\alpha_{oB}^{Nluc}$ (25 ng), $G\alpha_z^{Nluc}$ (25 ng), $^{Venus}G\gamma_2$ (50 ng), $^{Venus}G\gamma_8$ (50 ng) and $^{Venus}G\gamma_9$ (50 ng).

Primary cortical and hippocampal neuronal cultures were prepared from embryonic day 17.5 mice as previously described[85]. The cortex or hippocampi were digested with trypsin, and cells were seeded in neurobasal medium (Ref 21103049; Gibco) supplemented with 2% B27 (Ref 17504-044; Gibco), 4 mM GlutaMAX, 25 µM glutamic acid, 100 U/mL penicillin, 100 µg/mL streptomycin and 10% FBS in 96-well plates. After three days in culture (DIV3), the culture medium was supplemented with 5 µM cytosine β-D-arabinofuranoside hydrochloride (C6645; Sigma-Aldrich) and incubated overnight. Then, 75% of the medium was replaced by neurobasal medium supplemented with B27, GlutaMAX and antibiotics. Neurons were then transfected with $G\alpha_{i1}^{Nluc}$ or $G\alpha_{oA}^{Nluc}$ (50 ng/well), together with $^{Venus}G\gamma_2$ (50 ng/well) using Lipofectamine 2000 on DIV7. The ratio of DNA to Lipofectamine 2000 ratio was 1:2. BRET experiments were performed on DIV10 or DIV11.

## Fluorescent imaging

Cerebellar granule neurons (CGNs) and transfected HEK293 cells were fixed with 4% formaldehyde and blocked with 2% bovine serum albumin (BSA) and 0.1% Triton X-100 in phosphate-buffered saline (PBS). The CGNs were incubated with primary GFP antibody (1:200; ab1218, Abcam, Shanghai, China) at 4 °C overnight. After washing three times with PBS, the cells were incubated with secondary anti-mouse antibody Alexa Fluor® 488 AffiniPure Donkey Anti-Mouse IgG (H + L) (1:500, 715-545-150, Jackson ImmunoResearch, Shanghai, China) at 25 °C for 2 h. The CGNs were then stained with DAPI for 15 min. The cells were washed with PBS and mounted with FluorSave reagent (AR1109, Boster Biological Technology Co. Ltd., Wuhan, China). Images were obtained with an Olympus FV1000 laser scanning confocal microscope (60 x objective for HEK293 cells and 40x objective for CGNs, Olympus Corporation, Tokyo, Japan) equipped with appropriate fluorescence and filters (FITC: 488/530 nm; DAPI: 405/449 nm). The images were digitized and saved in TIFF format.

## BRET measurement

CGNs were starved in HEPES-buffered saline (HBS) containing 10 mM HEPES pH 7.4, 140 mM NaCl, 4 mM KCl, 2 mM MgSO$_4$, and 1 mM KH$_2$PO$_4$. Cortical and hippocampal neurons were starved with artificial cerebrospinal fluid (aCSF) buffer containing 140 mM NaCl, 2 mM CaCl$_2$,

3 mM KCl, 10 mM HEPES, and 10 mM D-glucose, at 37 °C for 1 h before the BRET experiments. BRET measurements were performed as previously described using the Mithras LB 940 multimode microplate reader (Berthold Technologies, Bad Wildbad, Germany)[49] with the programme MikroWin (Version 4.41) or PHERAstar FS (BMG Labtech, USA)[86] with the programme PHERAstar control (Version 4.00 R4). The signals emitted by the donor (460–500 nm band-pass filter, Em 480 nm) and the acceptor (510–550 nm band-pass filter, Em 530 nm) were recorded by Mithras LB 940 after the addition of 10 µM furimazine. All measurements were performed at 37 °C. The BRET signal was determined by calculating the ratio between the emission of acceptor and donor (Em 530 nm / Em 480 nm). The basal BRET ratio (BRET$_{basal}$) of cells was recorded before the stimulation with drugs or buffer. The change in BRET ratio (net BRET) was obtained by subtracting the BRET ratio after agonist treatment from the basal BRET (BRET$_{basal}$-BRET$_{agonist}$). Agonist-induced change in BRET ratio for the different G$_{i/o}$ sensors was expressed as percentage of the basal signal ((BRET$_{basal}$-BRET$_{agonist}$/BRET$_{basal}$) x100). To study the kinetics, the BRET was measured in real time with a counting time of 0.5 s, and the drugs were injected using the Mithras LB 940 injectors at the indicated time.

## Quantitative reverse transcription PCR

Total RNA was extracted using standard methods (Trizol, Invitrogen) from CGNs and HEK293 cells. Quantitative reverse transcription PCR (qRT-PCR) was carried out with SYBR Green (Vazyme Biotechnology, Nanjing, China) according to the manufacturer's protocol as reported[87]. *RplpO* was used as an internal reference for normalization, and the ΔΔCt method was adopted to analyse quantitative PCR data. The primer sets used were referred to previous references[75,88]. For mouse (CGNs), *Gnai1* (NM_010305), Forward (Fw): 5'-AAGCT-GACTCGCCTTCCCAG-3', Reverse (Rv): 5'-GTAGTTTACAGTTCTCCA-CACG-3'; *Gnai2* (NM_008138), Fw: 5'-TGCCTTGAGTGTGTCTGCGTG-3', Rv: 5'-CTCAGTGACGTTGGCAGTTG-3'; *Gnai3* (NM_010306), Fw: 5'-GTGCAGTCCGTGTACAAGAG-3', Rv: 5'-GATGAATGGATCCGAGCCAC-3'; *Gnao1* transcript isoform A (NM_010308) Fw: 5'-AGGAA-GACGGACTCCAAGATG-3', Rv: 5'-AGTCGAAGAGCATGAGAGAC-3'; *Gnao1* transcript isoform B (NM_001113384) Fw: 5'-AGGAA-GACGGACTCCAAGATG-3', Rv: 5'-AGATGTGTCTGTGAACCACTTG-3'; *Gnaz* (NM_010311), Fw: 5'-CAGCCGTGCTTAGAAACATCG-3', Rv: 5'-TCTAGTGACACTCCACCTCC-3' and *RplpO* (NM_007475), Fw: 5'-CTCACTGAGATTCGGGATATG-3', Rv: 5'-CTCCCACCTTGTCTCCAGTC-3'. For human (HEK293 cells), *GNAI1* (NM_002069), Fw: 5'-CATCTCT-GACCTTGTTTCAGC-3', Rv: 5'-CTTCAACCCAGTGACAACACG-3'; *GNAI2* (NM_002070), Fw: 5'-ACTCCGTGCCTTGAGTGTG-3', Rv:5'-TTGTCTGGAACAGCCCTTGG-3'; *GNAI3* (NM_010306), Fw: 5'-GGAAAGTTACGTTCACTTCAACC-3', Rv: 5'-TTGGACCCCAAAAGG-CACTG-3'; GNAO1 transcript isoform A (NM_020988) Fw: 5'-AGAAAGGCTGACGCCAAGAT-3', Rv: 5'-AGTCGAAGAGCATGAGAGAC-3'; *GNAO1* transcript isoform B (NM_138736) Fw: 5'-AGAAAGGCT-GACGCCAAGAT-3', Rv: 5'-TGGACGTGTCTGTGAACCAT-3'; *GNAZ* (NM_002073), Fw: 5'-CTACGAGGATAACCAGAC-3', Rv: 5'-TACGTGTTCTGGCCCTTG-3' and *RPLPO* (NM_053275), Fw: 5'-ATG-CAGCAGATCCGCATGT-3', Rv: 5'-TTGCGCATCATGGTGTTCTT-3'.

## Cell membrane preparations

The cell membranes were prepared as reported[89]. HEK293 cells or CGNs were washed three times with PBS, then scraped and collected by centrifugation at 170 g for 5 min. Cells were resuspended in 500 µl Tris buffer (50 mM Tris pH 7.4, 50 mM NaCl) with cOmplete protease inhibitor cocktail (Roche) and crushed through a 26 gauge 5/8 inch needle attached to a syringe for 30 passages. After centrifugation at 860 g at 4 °C for 5 min, liquid supernatants were transferred to high speed centrifuge tubes and centrifuged at 44000 g at 4 °C for 20 min by Optima MAX-TL ultracentrifuge (Beckman Coulter, Brea, CA, USA). The precipitated membranes were diluted gently in 50 µl Tris-NaCl

buffer. The protein amount was further determined using BCA protein assay kit and 20 µg protein was loaded for western blotting detection.

## Western blotting analysis

Cells were treated using RIPA lysis buffer (50 mM Tris pH 7.4, 150 mM NaCl, 1% NP-40, 0.5% sodium deoxycholate, 0.1% SDS, sodium ortho-vanadate, sodium fluoride, EDTA, leupeptin; Beyotime Bio., Cat. P0013C, China) and protein concentrations were determined using the BCA protein assay kit. Equal amounts of protein (20 µg) from total cell lysis or cell membranes were separated by SDS–polyacrylamide gel electrophoresis (PAGE) on 10 to 12% gels. Proteins were transferred to nitrocellulose membranes and washed in blocking buffer (5% nonfat dry milk in Tris-buffered saline and 0.1% Tween 20) for 2 h at 25 °C. The blots were incubated with the primary monoclonal antibodies anti-GB1 mAb (1:1000, ab55051, Abcam, Shanghai, China), anti-G$\beta_2$ rabbit (1:1000, A9643, ABclonal Technology, Wuhan, China), anti-β-actin (1:3000, KM9001T, Sungene Biotech., Tianjin, China), the rabbit polyclonal antibodies anti-CB1 (1:1000, A1447, ABclonal, Wuhan, China), anti-$\alpha_{2A}$AR (1:1000, A2809, ABclonal, Wuhan, China), anti-G$\beta_1$ (1:1000, A1867, ABclonal, Wuhan, China), anti-G$\beta_3$ (1:1000, A1387, ABclonal, Wuhan, China) and anti-G$\beta_5$ (1:1000, A4447, ABclonal, Wuhan, China). The primary antibodies were at the relevant dilution overnight at 4 °C and then incubated with DyLight 800 4 X PEG conjugated secondary antibodies (1:20,000, anti-mouse IgG, #5257; 1:20,000, anti-rabbit IgG, #5151, Cell Signaling Technology, Shanghai, China) for 2 h at 25 °C. Membranes were imaged using an Odyssey infrared scanner (LI-COR Biosciences, Lincoln, NE, USA) at 700 nm.

## Statistical analysis

Results are presented as the mean ± SEM of at least three independent experiments. Statistical analysis was performed using GraphPad Prism 9.5.1 software (GraphPad Software Inc., San Diego, CA, USA). Dose-response experiments were analysed using nonlinear curve fitting for the log (agonist) vs. response (three parameters) curves. Statistical analysis was performed using the Ordinary one-way ANOVA with a Dunnett's *post-hoc* multiple comparison test or unpaired *t*-test (two-tailed) or paired *t*-test. $P < 0.05$ was considered to be statistically significant.

## Reporting summary

Further information on research design is available in the Nature Portfolio Reporting Summary linked to this article.

# Data availability

All data generated in this study are provided in the main text, Supplementary information and source data files. The raw data and *p*-values for all Figures and Supplementary Figs. are available in Source Data file accompanying this paper. Source data are provided with this paper.

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

## Acknowledgements

We thank Dr Julie Perroy (IGF) for providing us Nluc cDNA. This work was supported by grants from the National Natural Science Foundation of China (NSFC) (grant number 32271198 to C.X.), the National Key R&D Programme of China (grant number 2022YFA1302901 to C.X.), the National Natural Science Foundation of China (NSFC) (grant number 32330049 and 82320108021 to J.L.), National Key R&D Programme of China (grant number 2022YFE0116600 and 2021ZD0203302 to J.L.), interdisciplinary Research Programme of HUST (grant number 2023JCYJ006 to C.X.), the Agence Nationale de la Recherche (ANR 18-CE11-0004-01 to J.-P.P.), and the Fondation Recherche Médicale (DEQ 20170336747 to J.-P.P and EQU202303016470 to P.R.). P.R. and J.-P.P. were supported by the Institut National de la Santé et de la Recherche Médicale (INSERM; International Research Programme «Brain Signal») and the Franco-Chinese Joint Scientific and Technological Commission (CoMix) from the French Embassy in China.

## Author contributions

C.X., J.L., P.R. and J.-P.P. designed the experiments. C.X. initiated the project, designed the sensors, set up the protocols and screened different agonists; C.X., Y.Z. and Y.L. detected the Gi/o protein response in HEK293 cells and CGNs; Y.L. optimized the transfection amount in HEK293 cells and prepared cell membranes; Y.L. and Y.Z. performed all the western blotting experiments; L.L. performed the dose-response in HEK293 cells treated with GABA_B receptor agonists; Y.L., L.L. and Z.X. produced constructs. P.L. and X.W. helped with the plasmid constructs and neuron preparations. C.X., Y.Z. and Y.L. collected and analysed the data. C.X., J.L., J.-P.P. and P.R. wrote the manuscript with input from all of the authors.

## Competing interests

P.R. and J.-P.P. are involved in a collaborative team between the CNRS and Cisbio Bioassays, Revvity group. The remaining authors declare no competing interests.
