## [Peer Review File · Nature Communications]

Specific pharmacological and Gi/o protein responses of some native GPCRs in neuronsREVIEWER COMMENTS

Reviewer #1 (Remarks to the Author):

Xu, Zhou and colleagues describe the use of bioluminescence resonance energy transfer (BRET) biosensors of heterotrimeric G protein dissociation as tools to investigate GPCR signaling in primary neurons. The study focuses on 5 related G proteins of the Gi/o family, which are investigated in HEK293 cells and primary neurons to compare different profiles of responses. After a series of experiments, the authors claim that they reveal specific properties of native GPCRs in neurons compared to heterologously expressed receptors in HEK293 cells, and that the approach they followed revealed ligand bias.

The general question of addressing GPCR / G protein signaling in their native environment is important. That the authors were able to detect GPCR responses in neurons has merit and is the main strength of the manuscript. However, neither the sensor design (see PMID: 34516756) or the ability to detect GPCR activity in their native context with G protein biosensors (see PMID: 35302493) are necessarily novel. Moreover, many of the claims about observing differences because the GPCR behaves differently in its native environment are inaccurate or false, and the authors do not take into account alternative interpretations for their results.

Next, I provide a more detailed description of specific points that are problematic.

1- Responses in neurons and HEK293 cells are different, but the claim that this is because GPCRs behave differently pharmacologically depending on the cell context is not well supported. The main item that is not controlled is receptor numbers in the two systems. Large differences are expected, and these differences in receptor numbers could account for the observed pharmacology.

2- The authors do not describe if the receptors studied in HEK293 are from the same species as the neurons. If they are not the same species, the comparison is not valid. Even if they are the same species, we do not know if they are comparing the same isoforms (e.g., different GABAB isoforms are going to be expressed in neurons but not cell lines).

3- The authors claim that they detect "ligand bias" in their studies comparing HEK293 and neurons. This is incorrect for most if not all the claims. The differences they observe correspond to system and/or measurement bias. To study ligand bias, the authors should report differences within a particular system to a reference ligand (see PMID: 35106752).

4- From results in Figure 5 and Table S2, the authors make qualitative claims about differences in efficacy for some compounds based on the G protein investigated in neurons, but that this is not seen in HEK293 cells. The first problem with this is that compound WIN55,211-2 and compound BAY5903074 do show differences in efficacy when comparing Gi vs Go even in HEK293. If there is a difference, it is quantitative and small, but no statistical measure is provided to support the claim. An issue that adds to this is that the pharmacological parameters (like efficacy) calculated for HEK293 cells are based on poorly fitted data because saturation is not reliably reached for all conditions (see Fig. 5a, c).

5- Another factor that could contribute greatly to the differences observed is the relative interference of G protein overexpression in different systems. It is not possible to know how much excess of exogenous G proteins is expressed in neurons, but this could account for the differences observed between neurons and cell line. Moreover, the endogenously expressed proteins in each system could also be competitors for the ones used as biosensors. In fact, this could explain why Go responses are better in HEK293 cells than in neurons. Go is very abundant in neurons and completely absent in HEK293 cells.

6- Related to the above, the responses detected in different systems could be due to the presence of different endogenous Gβ instead of different properties of the GPCRs per se as claimed by the authors.

7- Another limitation is that the biosensor system relies on one Gγ subunit, which may or may not be relevant for the particular system under study. It is known that different Gγ's can lead to marked differences in signaling (see PMID: 33667409).

8- The authors claim in the manuscript that they have developed an "optimized" sensor but the evidence for such optimization is absent. In figure S1 that should show that, as expected, Nluc provides higher counts, but then they do not prove that this leads to a better signal to noise when compared side by side to whatever their reference for optimization is (they even mention that it is "expected to provide much improved signal to noise", but it is not tested).

9- Discussion. The claim "Our biosensors are expected to minimally perturb the native situation" --- without evidence to support this expectation, the claim is not valid. One could expect that the biosensor will perturb native signaling without evidence otherwise.

10- Discussion. The claim "However, these biosensors were not compatible with the study of native receptors in primary neurons" is misleading. The biosensors they refer to might have not been tested in

neurons, but that does not mean that they were not compatible with the study of native receptors in neurons. In fact, this manuscript supports that they might be, since they are based on a similar principle.

Reviewer #2 (Remarks to the Author):

In the manuscript by Xu, Zhou, et al., the authors describe the use of bioluminescence resonance energy transfer (BRET)-based G protein dissociation biosensors to investigate the pharmacology of a subset of natively-expressed G protein-coupled receptors (GPCRs) in primary neurons. In doing so, the authors report divergences in the coupling and pharmacology of select receptors expressed natively in the neurons versus those transiently expressed in HEK293 cells, which are routinely used to determine ligand pharmacology. The premise of the paper is an important one for understanding receptor pharmacology work within relevant physiological contexts, which may help steer drug development and therapeutic application in addition to furthering basic cellular biology, but there are some concerns worth addressing – primarily related to biosensor utility and data interpretation – described below.

1. The authors describe the use of “optimized” BRET sensors for these studies, but use previously published insertion sites for the donor molecule, using a smaller NanoLuc instead of RLuc or RLuc8, and pair these with G α 2 fused to an acceptor mVenus with a short linker. While BRET is produced, and the authors do observe a change in BRET following drug administration for numerous receptors, there does not actually appear to have been much of an optimization process for these constructs, and the net BRET between vehicle and saturating doses of drug is relatively small for some receptors.
2. The use of G α 2 is not an inherent issue (though, as the authors mention, there is the possibility of the full trimer components impacting aspects of signaling, see Masuho et al., 2021), but it is unclear if these constructs as described are truly “optimal” for BRET, particularly for dynamic range – how, for example, do these sensors compare to the existing T sensors in, say, HEK293 cells? Optimization through insertion site sampling, G α swapping, linker optimization, etc. would have been particularly necessary using a donor molecule about half the size of standard luciferases where the orientation/spatial aspects contributing to BRET would likely be influenced. The advantages described with NanoLuc having a brighter intensity contributing to increased signal-to-noise would benefit the field if the constructs underwent further optimization or were shown, in fact, to be optimized.
3. On a related note, were the authors able to detect endogenous activation with their sensors in HEK293 cells? .

4. While TRUPATH and the predecessor dissociation constructs are not sensitive to receptor reserve, there is a minimal threshold of receptor expression necessary to distinguish signal from noise. The receptors tested in primary neurons here, particularly CB1, are highly enriched in neurons. Within the text, there seems to be a contradiction where it is stated that the inability to detect endogenous responses in HEK293s is a potential limiting factor of TRUPATH (see Introduction), but then saying that TRUPATH may serve as a better alternative for the applications here (see Discussion). Did the authors try using the TRUPATH sensors (even just the G α and G $\beta\gamma$ components) for comparison? This would be useful to know, since TRUPATH may in fact be useful for the application presented here (at least for certain receptors; see comment 4).

5. In the discussion, the authors bring up some important caveats regarding ligand versus location bias to describe their findings. A recent paper by Wall et al., for example, showed that the lack of expression of G α in post- versus pre-synaptic neurons contributes to the effects of an adenosine 1A receptor agonist's effects in vivo and in ex vivo slice physiology. In the case of this particular ligand, its effects appear to be both due to ligand bias (as this ligand is G α -biased in HEK293 cells) and location biased in the specific synapses under investigation in their ex vivo work. The author's findings in this work are both interesting and important, but I think their ability to interpret them could be bolstered by an additional set of experiments in which the localization of their G proteins are determined via immunocytochemistry against the NanoLuc (which an antibody exists) to determine where their transfected G proteins are expressed. Again, the authors do right in bringing this up in the discussion, but I think there is an opportunity to explore this more directly with reasonable feasibility. I recognize that similarly doing so for native (i.e., unmodified) receptors would be difficult given the lack of reliable antibodies, though there may be some information from the literature regarding their localization (dendrites, neurons, somas, etc.) that could be paired with the insights from G protein localization studies.

6. In Figure 3, the authors show that some receptors do not show detectable changes in BRET in primary neurons even in response to incredibly high concentrations of agonists, while others do (which they follow-up with concentration-response curves). Are these receptors known to be expressed in these particular neurons, and can the authors comment upon the lack of response? For example, is this due to low receptor density below which signal and noise can be discerned? This does not diminish the findings of the receptors that did respond and that were characterized, but it does emphasize some of the caveats and shortcomings described in the discussion, and the general applicability to the use of these or related biosensors in studying certain receptors within certain cellular contexts.

7. Some work on the writing/presentation would benefit the manuscript.

Reviewer #3 (Remarks to the Author):

The manuscript entitled „ Specific pharmacological and Gi/o protein coupling. Properties of native GPCRs in neurons” by Xu et al. reports about an interesting study on the G protein selectivity and agonist bias of certain endogenously expressed neuronal GPCRs in comparison to heterologously expressed GPCRs in HEK cells. The study is based on more sensitive BRET-based biosensors for G proteins for which the Ga subunit is fused to a NanoLUC and the Ggamma2 subunit carries a Venus. By relying on endogenous Gbeta the authors aim to study endogenous GPCR signaling on the level of G protein activation. For 3 endogenous GPCRs the authors were successful to measure G protein activation: alpha 2, CB1 and GABA-B receptors. Major findings were, that the G protein selectivity was quite different for heterologously expressed receptors in HEK cells compared to endogenously expressed receptors in primary neurons, specifically cerebellum granule neurons (CGNs).

In HEK cells the authors reported largest BRET amplitudes for G α oB for all three receptors. In CGNs both G α oA and G α oB showed the smaller BRET changes, even though they were expressed to similar levels to the other Ga subunits. Remarkably, the coupling profile of endogenous cannabinoid receptors as well as GABAB receptors were substantially different in neurons compared to HEK cells as baclofen was almost unable to activate Gai3 in neurons, but robustly activated it in HEK cells, and the CB1-agonist CP 55940 activated Gai1 and Gai3 to a similar extent in CB1-R expressing HEK cells, whereas the coupling to Gai1 was enhanced in CGNs. Interestingly the potency of 4 different CB1 Agonists to activate different G protein subtypes was greatly different in neurons compared to HEK cells. Very remarkable, for some agonists but not for others the potency was actually increased in CGNs compared to HEK cells, which is in contrast to the situation seen for GABAB-Rs and alpha2 receptors.

Overall, the study is timely and the results are highly important for the field of GPCR research. However, there are two main caveats (point 1 +2) that need careful attention in order to strengthen the conclusions. In addition, the study would benefit if the authors could show how much of the differential G protein selectivity in neurons is actually due to the endogenous expression level. Overexpression of the receptors in neurons and reduction of functional receptor expression in HEK cells could bring in some more light, to distinguish between effects of expression levels versus effects of the neuronal environment.

As this study in the present form does not address potential mechanisms, it would benefit if high resolution microscopy of the fluorescent G proteins would compare subcellular localization of the different G proteins in neurons.

1) The issue whether the G protein sensor reproduces the “endogenous” G protein coupling selectivity in neurons needs some more attention: The authors state:

In these experiments, the amount of G $\alpha\beta\gamma$ complexes that can produce a BRET signal with G α is expected to be limited by the endogenous level of G $\beta\gamma$ subunits that are able to form a complex with the G α Venus.

As the stability and lifetime of endogenous G α might very well depend on the expression levels of its lifelong partner G β , and even more, the expression level of G $\beta\gamma$ might depend on how well it finds its partner G α , it is of importance to actually experimentally check this assumption by testing the influence of G β -Venus on the expression level of endogenous G α and also the influence of coexpression of G β -subunits. Whereas this is an easy experiment in HEK cells, it might be difficult to assess in primary neurons, due to the low transfection efficiency, however I feel this is important.

2) More complicated, as the luminescent and fluorescent G protein subunits are expressed on top of the endogenously expressed G proteins, it remains unclear how those would actually interfere with the coupling selectivity. This issue is probably much more difficult to experimentally address. One solution could be the use of selective G α -KO mice to see whether for instance a selective G α -KO would change the selectivity profile of endogenous receptors measured with the novel BRET sensors presented here.

Specific comments:

A) As the degree to which G proteins get activated by the constitutive activity of GPCRs greatly depends on the expression level of GPCRs the results concerning the basal activity of GABAB receptors are not really conclusive unless expression level independent differences would be observed. Therefore, these results should be either taken out or should go into the supplemental information.

B) Line 745: ...CB1 receptor,

CB1 receptor doesn't make sense to me- should be alpha2A?

C) These data suggest that the GABAB receptor has similar properties for G i/o coupling in the different neurons, while this coupling is stronger in the HEK293 cells.

Not the coupling might be stronger in HEK cells but rather the amount of receptor expression, which gives rise to the spare receptor phenomena.

D) A direct comparison HEK versus CGN for Rac BHFF with the identical G protein assay would have been nice. Also experimental evidence to test how much the overexpression of receptors may be helpful for interpretation of the observed effects.

Reviewer #4 (Remarks to the Author):

This manuscript uses slightly modified, well-characterised BRET-based biosensors to investigate the Gi/o protein signalling of three native neuronal GPCRs: GABAB, CB1 cannabinoid, and $\alpha 2$ adrenoceptors. Comparisons were also made between the primary neurons (taken from mice) and the HEK 293 cell line. The authors have concluded that native neuronal receptors show different efficacy and potency responses to several ligands when compared to the HEK 293 recombinant system. The take-home message is that it is important to characterise new drugs in both recombinant and primary systems to ensure correct mechanistical insight into the effects.

The use and transfection of primary neurons in vitro is commendable as these are tricky cells to work with, and the biosensors work very well to produce clear results with good error margins. Their use of overexpressing the alpha and gamma subunits without the beta is an inspired choice, as three-plasmid transfection would not have yielded as good results. However, there are also other methods available, which I will detail below. I agree with the authors that the native receptor population is an important and sometimes overlooked aspect of drug development, and I like that the discussion addresses the fact that different interacting proteins will be present in different cell types. The results gathered in this paper support their stance on their conclusions. The introduction has reviewed a good cross-section of the literature. Although the biosensors themselves are not novel, their use in neuronal cells is novel and important. The receptors investigated also represent a good cross-section of druggable disease targets. The authors have also used a good variety of ligands to characterise the receptor G protein response.

Questions and constructive feedback for the authors:

Major points

1. The major caveat that is not addressed throughout the whole paper is the difference in GABA receptor expression between the two cell types (CGNs and HEK cells). In line 164, the authors seem surprised that there is an increase in drug potency in the HEK cells. This is stated in lines 167-9 that neuronal receptors don't couple to Gi/o proteins as readily as HEK cells. Again, this is repeated in lines 181-3. In lines 190-2, the authors note that the constitutive response is not present in neurons – again this is probably due to a lower receptor density. Lines 335-7 fail to address the receptor density, although it is also true that a difference in cellular environment may cause changes in ligand effects. Reading the methods, it seems that the GABA receptors transfected into the HEK cells are under a CMV promotor in the pRK5 plasmid vector. This will cause receptor overexpression, which causes increase in potency due to the law of mass action (H J Motulsky and L C Mahan, *Molecular Pharmacology* 1984, 25 (1) 1-9). I suggest that the authors need to address this shortcoming by measuring receptor density in the two systems. Radioligand binding could be used for this, or, if a suitable antibody or fluorescent ligand is available, then ELISA, Western blotting and/or immunochemistry.
2. In the methods (lines 337-9) it is not clear which species of receptors are being transfected into the recombinant HEK cell line. In the primary neurons, the receptors will be of mouse origin, however in the

cell lines this may be human – the NM database number for the receptor sequence is not given. I am not sure how conserved the receptors are between species, but this is an important point the authors need to address as differences in ligand potency with the biosensors could be due to a species difference in receptor.

3. Although the English is simple enough for a wide audience and the meanings are clear and concise on the whole, the grammar is not always at a good standard. For example, the first sentence (lines 40 and 41) reads: G protein-coupled receptors (GPCRs) ubiquitously expressed in every cell types, constitute the largest family of membrane proteins. This should be: G protein-coupled receptors (GPCRs) ARE ubiquitously expressed in every cell TYPE, AND constitute the largest family of membrane proteins, or, G protein-coupled receptors (GPCRs), ubiquitously expressed in every cell TYPE, constitute the largest family of membrane proteins. I will not point out all grammatical errors here, but I suggest the authors get an external person to check the English grammar prior to submission in the future.

Minor points

4. In the abstract, GPCRs are classed as ‘the most important drug targets’ (line 27). Although I am a GPCR researcher and agree, I think researchers in other fields may disagree. Perhaps something more neutral, such as ‘are the most targeted by currently registered drugs on the market’?

5. I like that fact that all subtypes of Gi/o proteins were investigated (line 93). However, why was Gz omitted? This is another crucial member of the inhibitory G protein family.

6. Lines 102-3 do not state the animal model which the neurons were taken from. This is important information for the reader.

7. Lines 104-5 state that in vitro culturing of cerebellum granule neurons are ‘a good model for the study of cellular and molecular mechanisms of survival and neuroprotection’. There is no reference to support this, or any reason given as to why this is the case, or why this is important to this paper.

8. Lines 115-20 I appreciate the comparison between the luciferases, however the authors may have missed a crucial reference where NanoLuciferase has already been inserted into these sensors previously: Schihada et al. 2021, Sci Signal DOI:10.1126/scisignal.abf1653, and/or Schihada et al. 2022, JBC DOI: 10.1016/j.jbc.2022.102328.

9. Line 129 has an approximate measure of transfection of the cerebellar neurons. How was this counted? How can they be sure that all cells with the Venus tagged subunit also express the NanoLuc tagged alpha subunit? Did the authors consider a single plasmid transfection with a 2A cleavage site or IRES two-gene expression to ensure that cell transfection would produce both subunits?

10. Line 130 states that the low level of transfection is normal, which is true. However, it is not addressed that any transfected cell will be ‘overexpressing’ the sensors through the plasmid promoter.

11. Lines 138-9 states that the G proteins are ‘barely activated’ in CGNs compared to HEK cells. This is a subjective statement. How much difference was there in potency and efficacy? Or were no suitable concentration response curves possible with the CGNs?

12. Lines 140-2 state that it is not the difference in sensor alpha subunits causing the difference seen between the two cell lines. Although this is true, I am wondering about the impact of beta subunit

expression between the two systems? This is not controlled, as the native beta units are used in the sensor.

13. Lines 179-81 - I don't understand this sentence. Are the authors trying to say that Rac BHFF can cause G protein activation and calcium mobilization by itself in other systems? It is not a purely allosteric ligand?

14. Do the authors mean 'expressed' rather than 'exhibited' in line 185?

15. Line 190 – do the authors mean that the inverse agonist is causing the receptor to be in its inactive conformation?

16. Lines 193-4 state that there is a difference in potency between the CGNs and HEK cells. Perhaps it should be addressed in the discussion that other endogenous receptors in the two cell types may impact these results.

17. Is the agonist Win 55,211-2, described in line 238, a super agonist?

18. I feel that the Emax does not need to be reported to two decimal places (lines 240-1). The span of the concentration response curve fit is probably not this accurate. One or no decimal places would be more precise.

19. Line 272 uses the word 'whatever'. This is used as a slang word in this context and is not suitable for a scientific paper. It is used again in line 354.

20. Line 273 states that other sensors will be developed for the other G proteins. I refer back to the paper I pointed out where they have inserted the NanoLuc into many different G protein alpha subunits (Schihada et al. 2021, Sci Signal).

21. In the discussion lines 312-3, it is stated that the poor coupling to Go proteins needs to be clarified. I would argue that this is not a problem as all receptors are different. I have found that some dopamine receptors prefer the Go subtype over Gi, and NPY receptors also (CNS expressed). Perhaps the receptors addressed in the paper are just not those associated with Go?

REVIEWER COMMENTS

Reviewer #1 (Remarks to the Author):

Xu, Zhou and colleagues describe the use of bioluminescence resonance energy transfer (BRET) biosensors of heterotrimeric G protein dissociation as tools to investigate GPCR signaling in primary neurons. The study focuses on 5 related G proteins of the Gi/o family, which are investigated in HEK293 cells and primary neurons to compare different profiles of responses. After a series of experiments, the authors claim that they reveal specific properties of native GPCRs in neurons compared to heterologously expressed receptors in HEK293 cells, and that the approach they followed revealed ligand bias.

The general question of addressing GPCR / G protein signaling in their native environment is important. That the authors were able to detect GPCR responses in neurons has merit and is the main strength of the manuscript. However, neither the sensor design (see PMID: 34516756) or the ability to detect GPCR activity in their native context with G protein biosensors (see PMID: 35302493) are necessarily novel. Moreover, many of the claims about observing differences because the GPCR behaves different in its native environment are **inaccurate or false**, and the **authors do not take into account alternative interpretations for their results**.

We thank the referee for highlighting the importance of measuring endogenous GPCR activation in primary neurons. The referee is right that the principle of our G protein sensors is not novel, as it simply represents an improvement of already described sensors using Nanoluc, instead of Rluc. The group of Schulte did indeed publish similar sensors also using Nanoluc, but these studies were published after we wrote our manuscript. These publications are now cited in our revised paper. Although we agree that other sensors revealed the activity of endogenous GPCRs and native G proteins as illustrated in M Bouvier's study, still there is no demonstration that this approach can be used with primary neurons. We still think our study being the first to succeed in getting pharmacological data with native, endogenous GPCRs in primary neurons in a throughput format (microplates) is a major achievement.

For the other points raised in this general statement, responses can be found below, in our answers to the specific points.

Next, I provide a more detailed description of **specific points that are problematic**.

1) Responses in neurons and HEK293 cells are different, but the claim that this is because GPCRs behave differently pharmacologically depending on the cell context is not well supported. The main item that is not controlled is **receptor numbers in the two systems**. Large differences are expected, and this **differences in receptor numbers could account for the observed pharmacology**.

We agree with the referee that higher receptor expression levels in transfected cells compared to the native receptors in neurons is expected to result in an increased agonist potency. If all agonists are full, the change in EC_{50} is expected to be the same for all ligands. Interestingly, this is not the case, for example Baclofen is out of such correlation for the GABA_B receptor (Fig. 2d, Supp Fig. 3d and f). Moreover, for the CB1 ligands, a higher potency is observed with the native receptors in neurons compared to the transfected receptors in HEK293 cells.

In the revised version, we better controlled the number of the three receptors investigated by quantifying the amount of these receptors using membrane preparations by western blot (see new Fig. 1i, Fig. 4d and Fig. 5a). In these experiments same amounts of membranes from transfected HEK293 cells and neurons were analyzed. Interestingly, while GABA_B and alpha2A receptors were 5 and 40 times more expressed in transfected HEK293 cells compared to neurons, CB1 receptor has a similar expression in the two systems.

2) The authors do not describe if the **receptors studied in HEK293 are from the same species** as the neurons. **If they are not the same species, the comparison is not valid.** Even if they are the same species, we do not know **if they are comparing the same isoforms** (eg., different GABA_B isoforms are going to be expressed in neurons but not cell lines).

This is a very good point raised by the referee. Our actual data were obtained using the mouse version of the CB1, α2A receptors and the rat version of the GABA_B receptor, such that they can be compared with data obtained with the primary neurons prepared from mice. Of note the rat GABA_B receptor sequences differ from mouse GABA_B by only two residues in GB1, and one residue in GB2 neither located close to the binding site nor within regions involved in the activation process.

3) The authors claim that they detect "ligand bias" in their studies comparing HEK293 and neurons. **This is incorrect for most if not all the claims. The differences they observe correspond to system and/or measurement bias.** To study ligand bias, the authors should report differences within a particular system to a reference ligand (see PMID: 35106752).

The referee is right, we removed any reference to biased signaling when we compare HEK293 and neurons, in our revised version of the manuscript.

4) From results in Figure 5 and Table S2, the authors **make qualitative claims about differences in efficacy for some compounds** based on the G protein investigated in neurons, but that this is not seen in HEK293 cells. The first problem with this is that compound WIN55,211-2 and compound BAY5903074 do show differences in efficacy when comparing Gi vs Go even in HEK293. If there is a difference, it is quantitative and small, but no statistical measure is provided to support the claim. An issue that adds to this is that the pharmacological parameters (like efficacy) calculated for HEK293 cells are based on poorly fitted data because saturation is not reliably reached for all conditions (see Fig. 5a, c).

Again, we thank the referee for pointing this out. Our initial statement was not justified, and then removed in the revised version of our manuscript.

5) Another factor that could contribute greatly to the differences observed **is the relative interference of G protein overexpression in different systems.** It is not possible to know how much excess of exogenous G proteins is expressed in neurons, but this could account for the differences observed between neurons and cell line. Moreover, the endogenously expressed proteins in each system could also be competitors for the ones used as biosensors. In fact, this could explain why Go responses are better in HEK293 cells than in neurons. **Go is very abundant in neurons and complete absent in HEK293 cells.**

We would like to stress that we did not transfect the β subunits in neurons, such that the amounts of heterotrimeric complexes are limited by the amount of endogenous β subunits. As such, any G protein sensor complexes formed is detrimental to the amounts of endogenous complexes. However, it is correct to assume that endogenous G proteins may limit the amounts of sensors being activated. We performed qPCR experiments to compare the relative expression of the 5 G protein α subunits in HEK cells, and in neurons. We did observe a large amount of Gi2 mRNA in HEK cells, while GoA mRNA is the major form in granule neurons (see new Fig. 1j).

Of note, we realized after submission of our initial manuscript that our Gi3 and GoB constructs were indeed mutated. As illustrated in our revised manuscript, we can indeed measure a nice activation of Gi3 and GoB proteins in neurons. Note that we systematically observed very similar variations in BRET signal for both GoA and GoB sensors, whereas only GoA is highly expressed in neurons. This brings further arguments that the endogenous G proteins do not affect much our sensor signals. It was mentioned in the revised version.

6) Related to the above, the responses detected in different systems **could be due to the presence of different endogenous Gbeta** instead of different properties of the GPCRs per se as claimed by the authors.

Yes, we fully agree that some of the differences can be due to the endogenous G β subunits. This was part of our proposal that proteins that can be part of the RG complex can be different between neurons and HEK cells providing a possible explanation for the differences observed. We now specifically mention the G β subunits as a possibility. We also compared the expression of different endogenous G β subunits (except G β 4 since our antibody against this subunit is of poor quality) in both neurons and HEK (see new Supp Fig. 1a).

7) Another limitation is that the biosensor system relies on one Ggamma subunit, which may or may not be relevant for the particular system under study. **It is known that different Ggamma's can lead to marked differences in signaling** (see PMID: 33667409).

As for the β subunits, the γ subunits can also influence the capacity of the GPCR to induce specific conformational changes in the G protein heterocomplex. It was now clearly stated in the Results and Discussion and this Martemyanov lab article was included. We indeed tested this possibility comparing the data obtained with γ 2, γ 8 and γ 9, and illustrate that indeed differences can be found depending on the G γ subunit being used. These data are shown in the new Figure 6.

8) The authors claim in the manuscript that they have developed an "optimized" sensor but **the evidence for such optimization is absent**. In figure S1 that should show that, as expected, Nluc provides higher counts, but then they do not prove that this leads to a better signal to noise when compared **side by side to whatever their reference for optimization is** (they even mention that it is *expected to provide much improved signal to noise*, but it is not tested). **The referee is right, we did not really "optimize" our sensor (this word was removed in the revised manuscript). We just generated a commonly used G protein sensor based on BRET signal between the G α and the Ggamma subunits, but using the Nanoluc rather than Rluc. We found this to surprisingly work well in neurons, but there has been no specific optimization. This point is now clear in the revised manuscript, specifying that changing Rluc to Nanoluc was sufficient to get reproducible and significant BRET variations in neurons, compared to the commonly used Rluc8 sensor. The side-by-side comparison is included in the new Supp Fig. 1d. We also compared our constructs with the TRUPATH ones (New Supp Fig. 1c).**

9) Discussion. The claim "Our biosensors are expected to minimally perturb the native situation" --- **without evidence to support this expectation, the claim is not valid**. One could expect that the biosensor will perturb native signaling without evidence otherwise. **This claim has been removed.**

10) Discussion. The claim "However, these biosensors were not compatible with the study of native receptors in primary neurons" is misleading. **The biosensors they refer to might have not been tested in neurons**, but that does not mean that they were not compatible with the study of native receptors in neurons. **In fact, this manuscript supports that they might be, since they are based on a similar principle.** **This sentence has been modified as follows: "However, these biosensors have never been tested in primary neurons."**

Reviewer #2 (Remarks to the Author):

In the manuscript by Xu, Zhou, et al., the authors describe the use of bioluminescence resonance energy transfer (BRET)-based G protein dissociation biosensors to investigate the pharmacology of a subset of natively-expressed G protein-coupled receptors (GPCRs) in primary neurons. In doing so, the authors report divergences in the coupling and pharmacology of select receptors expressed natively in the neurons versus those transiently expressed in HEK293 cells, which are routinely used to determine ligand pharmacology. **The premise of the paper is an important one for understanding receptor pharmacology work within relevant physiological contexts**, which may help steer drug development and therapeutic

application in addition to furthering basic cellular biology, but there are **some concerns worth addressing** – primarily related to biosensor utility and data interpretation – described below. We thank the referee for highlighting the importance of analyzing the pharmacological and signaling properties of endogenous GPCRs in a physiological context. We agree that having access to a suitable approach to do so will be beneficial for the drug development programs.

1) The authors describe the use of “optimized” BRET sensors for these studies, but use previously published insertion sites for the donor molecule, using a smaller NanoLuc instead of RLuc or RLuc8, and pair these with G γ 2 fused to an acceptor mVenus with a short linker. While BRET is produced, and the authors do observe a change in BRET following drug administration for numerous receptors, **there does not actually appear to have been much of an optimization process for these constructs**, and the net BRET between vehicle and saturating doses of drug is relatively small for some receptors.

This point was also raised by referee #1 (see his/her point #8). The word “optimized” was removed in the revised manuscript. As indicated above, our G protein biosensors are not original, as they were made more sensitive thanks to the use of NLuc. However, we show that such a modification of these biosensors make them suitable to be used in a microplate format to measure endogenous GPCR activity in primary neurons. This has never been validated before, and we show these sensors allow both the pharmacological characterization of native GPCRs and the analysis of their G protein coupling specificity. A side-by-side comparison of data obtained with the NLuc and RLuc sensors is shown in Supp Fig. 1, including a comparison of our constructs with the TRUPATH ones (New Supp Fig. 1c).

It is true that the change in BRET can be small, but still significant and reproducible.

2) The use of G γ 2 is not an inherent issue (though, as the authors mention, there is the possibility of the full trimer components impacting aspects of signaling, see Masuho et al., 2021), but it is unclear if these constructs as described are truly “optimal” for BRET, particularly for dynamic range – how, for example, do these sensors compare to the existing T sensors in, say, HEK293 cells? **Optimization through insertion site sampling, G γ swapping, linker optimization, etc. would have been particularly necessary using a donor molecule about half the size of standard luciferases where the orientation/spatial aspects contributing to BRET would likely be influenced.** The advantages described with NanoLuc having a brighter intensity contributing to increased signal-to-noise would benefit the field if the constructs underwent further optimization or were shown, in fact, to be optimized.

We agree that our G protein sensors could be further optimized to get better signal to noise ratios, and even increase the sensitivity in order to detect the activity of GPCRs natively expressed at low levels, including in HEK293 (see point #3 below). However, as our sensors were already good enough to start comparing data in native cells like neurons in a throughput format, allowing a comparison with what can be obtained in transfected HEK cells, we decided to first see whether differences could be observed to demonstrate how important it is to be able to work with native receptors. We think our work will stimulate us and others to further improve such biosensors, including the TRUPATH sensors, as we illustrate here the utility of doing so.

As indicated above, and in our answer to the point #8 of the first referee, we are now much more clear on the type of biosensors we used (see new Fig.1c and Supp Fig.1c), not indicating that we “optimized” or “developed” new biosensors.

Regarding the use of G γ 2, and as requested by the other referees, we now include new data using other G γ (γ 8 and γ 9), and found important differences in the ability of native GPCRs to activate specific G protein subtypes, depending of the G γ subunit present in the G protein heterotrimer (see new Fig. 6).

3) On a related note, were the authors able to detect endogenous activation with their sensors in HEK293 cells?

This is an interesting point raised by the reviewer. Interestingly, we show that our biosensors are able to report the activation of endogenous LPA receptor in HEK293 cells, as reported by the Schulte group (Schihada et al 2021, Sci Signaling). These data are now shown in the new Supp Fig. 2a-b.

4) While TRUPATH and the predecessor dissociation constructs are not sensitive to receptor reserve, there is a minimal threshold of receptor expression necessary to distinguish signal from noise. The receptors tested in primary neurons here, particularly CB1, are highly enriched in neurons. Within the text, there seems to be a contradiction where it is stated that the inability to detect endogenous responses in HEKs is a potential limiting factor of TRUPATH (see Introduction), but then saying that TRUPATH may serve as a better alternative for the applications here (see Discussion). Did the authors try using the TRUPATH sensors (even just the $G\alpha$ and $G\gamma$ components) for comparison? This would be useful to know, since TRUPATH may in fact be useful for the application presented here (at least for certain receptors; see comment 4).

The referee is right. Our presentation and discussion about the TRUPATH approach have been modified. Also, we did add a comparison between our Nluc sensors and the equivalent Rluc sensors in HEK cells that are quite similar (G_{i1} , $i2$, $i3$) or identical (G_{oA} and G_{oB}) to those of the TRUPATH sensors (Supp Fig. 1c). We did not succeed in getting significant results with the TRUPATH sensors in primary neurons (data not shown).

5) In the discussion, the authors bring up some important caveats regarding ligand versus location bias to describe their findings. A recent paper by Wall et al., for example, showed that the lack of expression of $G\alpha_{ob}$ in post- versus pre-synaptic neurons contributes to the effects of an adenosine 1A receptor agonist's effects in vivo and in ex vivo slice physiology. In the case of this particular ligand, its effects appear to be both due to ligand bias (as this ligand is $G\alpha_{ob}$ -biased in HEK293 cells) and location biased in the specific synapses under investigation in their ex vivo work. The author's findings in this work are both interesting and important, but I think their ability to interpret them could be bolstered by an additional set of experiments in which the localization of their G proteins are determined via immunocytochemistry against the NanoLuc (which an antibody exists) to determine where their transfected G proteins are expressed. Again, the authors do right in bringing this up in the discussion, but I think there is an opportunity to explore this more directly with reasonable feasibility. I recognize that similarly doing so for native (i.e., unmodified) receptors would be difficult given the lack of reliable antibodies, though there may be some information from the literature regarding their localization (dendrites, neurons, somas, etc.) that could be paired with the insights from G protein localization studies.

We thank the referee for this comment. Indeed, we agree that getting more information on the subcellular localization of GPCRs and G proteins, especially in neurons, would be very informative. Getting such information would certainly be feasible for the G proteins, but would not be informative without a parallel localization of the receptor. As such, we think such experiments are not needed for the present study that already shows a large number of data, and will be more appropriate for a follow up study.

6) In Figure 3, the authors show that some receptors do not show detectable changes in BRET in primary neurons even in response to incredibly high concentrations of agonists, while others do (which they follow-up with concentration-response curves). **Are these receptors known to be expressed in these particular neurons, and can the authors comment upon the lack of response?** For example, is this due to low receptor density below which signal and noise can be discerned? This does not diminish the findings of the receptors that did respond and that were characterized, but it does emphasize some of the caveats and shortcomings described in the discussion, and the general applicability to the use of these or related biosensors in studying certain receptors within certain cellular contexts.

We examined the effect of a few agonists acting at G_i -coupled GPCRs in granule neurons: the PAF, DOR-MOR, 5HT1B/D, α_2AR , M2/4R, D2/4R, CB1 and $GABA_B$, all expressed in these

neurons at this development stage (Maurel et al., 2011 PMID 21575240). Among these, the agonists activating α_2 AR, CB1, M2/4R, D2/4R and GABA_B could nicely activate the G_{i1} and G_{oA} sensors. These receptors correspond to those for which higher amount of mRNA were detected in these cultured neurons (Maurel et al., BMC Gen 2011 PMID 21575240). So, yes, the expression level of the native receptors appears critical to get significant responses. Our data may stimulate further optimization of such G protein biosensors to detect the activity of GPCRs expressed at a lower level.

7) Some work on the writing/presentation would benefit the manuscript. The manuscript has been checked carefully to improve its readability. It has also been checked for the English.

Reviewer #3 (Remarks to the Author):

The manuscript entitled „ Specific pharmacological and Gi/o protein coupling. Properties of native GPCRs in neurons” by. Xu et al. reports about an interesting study on the G protein selectivity and agonist bias of certain endogenously expressed neuronal GPCRs in comparison to heterologously expressed GPCRs in HEK cells. The study is based on more sensitive BRET-based biosensors for G proteins for which the Ga subunit is fused to a NanoLUC and the Ggamma2 subunit carries a Venus. By relying on endogenous Gbeta the authors aim to study endogenous GPCR signaling on the level of G protein activation. For 3 endogenous GPCRs the authors were successful to measure G protein activation: alpha 2, CB1 and GABA-B receptors. Major findings were, that the G protein selectivity was quite different for heterologously expressed receptors in HEK cells compared to endogenously expressed receptors in primary neurons, specifically cerebellum granule neurons (CGNs).

In HEK cells the authors reported largest BRET amplitudes for G α B for all three receptors. In CGNs both G α A and G α B showed the smaller BRET changes, even though they were expressed to similar levels to the other Ga subunits. Remarkably, the coupling profile of endogenous cannabinoid receptors as well as GABAB receptors were substantially different in neurons compared to HEK cells as baclofen was almost unable to activate Gai3 in neurons, but robustly activated it in HEK cells, and the CB1-agonist CP 55940 activated Gai1 and Gai3 to a similar extent in CB1-R expressing HEK cells, whereas the coupling to Gai1 was enhanced in CGNs. Interestingly the potency of 4 different CB1 Agonists to activate different G protein subtypes was greatly different in neurons compared to HEK cells. Very remarkable, for some agonists but not for others the potency was actually increased in CGNs compared to HEK cells, which is in contrast to the situation seen for GABAB-Rs and alpha2 receptors.

We should point out that we realized after submission of this story that two of our constructs (Gi3 and GoB) were indeed carrying a point mutation. Experiments were then repeated with validated constructs and the difference between the Go netBRET signals between HEK and neurons is smaller, but still clear for CB1 (Fig. 5 and Fig. 7).

Overall, the study is timely and the results are highly important for the field of GPCR research. However, there are two main caveats (point 1 +2) that need careful attention in order to strengthen the conclusions. In addition, the study would benefit if the authors could show how much of the differential G protein selectivity in neurons is actually due to the endogenous expression level. Overexpression of the receptors in neurons and reduction of functional receptor expression in HEK cells could bring in some more light, to distinguish between effects of expression levels versus effects of the neuronal environment. As this study in the present form does not address potential mechanisms, it would benefit if high resolution microscopy of the fluorescent G proteins would compare subcellular localization of the different G proteins in neurons.

As suggested by the referee, we examined whether the over-expression of CB1 in neurons would affect the G protein coupling profile. As illustrated in Supp Fig. 5c-d, the transfection of

CB1 in neurons did not change the G protein sensor signals for any of the sensors tested, and did not affect the expression level of these sensors. We did analyze the expression level of CB1 by western blot, and did observe a significant higher level of CB1 in transfected cultures, despite the very low transfection efficiency, indicating that in the transfected neurons also expressing the sensors, the level of CB1 might be really high. Indeed, as asked by referee #1 (point #1), CB1 receptor has a similar expression in transfected HEK293 cells and in neurons as observed in western blot (see new Fig. 5a).

In the revised version, we also better controlled the number of the two other receptors, GABA_B and alpha2A, by quantifying the amount of these receptors using membrane preparations by western blot (see new Fig. 1i and Fig. 4d). In these experiments same amounts of membranes from transfected HEK293 cells and neurons were analyzed. Interestingly, in contrast to CB1, GABA_B and alpha2A receptors were 5 and 40 times more expressed in transfected HEK293 cells compared to neurons.

Regarding the sub-localization of the G protein subunits in neurons, although such information will be interesting, we think it won't bring essential information in the absence of the parallel localization of the studied GPCRs, as explained above to the referee #2 (point #5).

1) The issue whether the G protein sensor reproduces the “endogenous” G protein coupling selectivity in neurons needs some more attention: The authors state: In these experiments, the amount of Gβγ complexes that can produce a BRET signal with Gα is expected to be limited by the endogenous level of Gβ subunits that are able to form a complex with the GγVenus.

As the stability and lifetime of endogenous Gβ might very well depend on the expression levels of its lifelong partner Gγ, and even more, the expression level of Gβγ might depend on how well it finds its partner Gα, it is of importance **to actually experimentally check this assumption by testing the influence of Gγ-Venus on the expression level of endogenous Gβ and also the influence of coexpression of Gα-subunits**. Whereas this is an easy experiment in HEK cells, it might be difficult to assess in primary neurons, due to the low transfection efficiency, however I feel this is important.

The referee is correct that the over-expression of both Gα and Gγ may stabilize Gβ, leading to a higher expression of the G protein. We then modified our text to mention this possibility. Moreover, we did test the effect of the over-expression of both Gα and Gγ on the expression level of Gβ1, β2, β3 and β5 in HEK293 cells (Supp Fig.1e). We also did this experiment in primary neurons where we did not see any change but this was expected because less than 5% of the neurons are indeed transfected.

2) More complicated, as the luminescent and fluorescent G protein subunits are expressed on top of the endogenously expressed G proteins, **it remains unclear how those would actually interfere with the coupling selectivity**. This issue is probably much more difficult to experimentally address. One solution could be **the use of selective Ga-KO mice** to see whether for instance a selective Gαo-KO would change the selectivity profile of endogenous receptors measured with the novel BRET sensors presented here.

The referee is correct that the presence of the endogenous, not labelled G proteins may affect the signal of the sensor through direct competition on the activated GPCR. This point was already raised by referee #1 (point #5), and we mention this possibility in our manuscript. However, it is interesting to note that we get the same effects on both GoA and GoB sensors in primary granule neurons that exclusively express GoA, suggesting that the presence of GoA did not influence the response generated by our sensor. It would have been interesting to perform experiments with neurons prepared from Gα-KO mice. However, we do not have access to these mice in our animal facilities (both in Montpellier and in Wuhan) and getting these animals would need more than a year, making this specific experiment not feasible in the due time of a paper revision.

Specific comments:

A) As the degree to which G proteins get activated by the constitutive activity of GPCRs greatly depends on the expression level of GPCRs the results concerning the basal activity of GABAB receptors are not really conclusive unless expression level independent differences would be observed. Therefore, these results should be either taken out or should go into the supplemental information.

The referee is right. The absence of observed constitutive activity of the GABA_B receptor in neurons might due to a lower expression compared to the transfected HEK. Indeed, our new data show a five time less amount of GABA_B in neurons than in transfected HEK. In addition, our biosensors could be not enough sensitive to detect this constitutive activity. This is why the data on the constitutive activity of GABA_B is now in Supp Fig. 3k.

B) Line 745: ...CB1 receptor,

CB1 receptor doesn't make sense to me- should be alpha2A?

Thanks for pointing out this mistake that has now been corrected.

C) These data suggest that the GABAB receptor has similar properties for Gi/o coupling in the different neurons, while this coupling is stronger in the HEK293 cells. Not the coupling might be stronger in HEK cells but rather the amount of receptor expression, which gives rise to the spare receptor phenomena.

This is indeed the most likely explanation. This is now clearly stated in our revised manuscript.

D) A direct comparison HEK versus CGN for Rac BHFF with the identical G protein assay would have been nice. Also experimental evidence to test how much the overexpression of receptors may be helpful for interpretation of the observed effects.

We did test Rac-BHFF in parallel in transfected HEK293 cells and in neurons. As illustrated in Fig 2e, Rac-BHFF has a clear ago-PAM activity in HEK cells, but its agonist activity is not detected in neurons. This can be explained by a higher expression level of the GABA_B receptor in HEK cells leading to receptor reserve compared to neurons. However, other explanations are possible, especially considering that the native GABA_B receptor are indeed associated to KCTDs. Clarification of this issue will need further experiments not in the scope of the present study.

Reviewer #4 (Remarks to the Author):

This manuscript uses slightly modified, well-characterised BRET-based biosensors to investigate the Gi/o protein signalling of three native neuronal GPCRs: GABAB, CB1 cannabinoid, and $\alpha 2$ adrenoceptors. Comparisons were also made between the primary neurons (taken from mice) and the HEK 293 cell line. The authors have concluded that native neuronal receptors show different efficacy and potency responses to several ligands when compared to the HEK 293 recombinant system. The take-home message is that it is important to characterise new drugs in both recombinant and primary systems to ensure correct mechanistical insight into the effects.

The use and transfection of primary neurons in vitro is commendable as these are tricky cells to work with, and the biosensors work very well to produce clear results with good error margins. Their use of overexpressing the alpha and gamma subunits without the beta is an inspired choice, as three-plasmid transfection would not have yielded as good results. However, there are also other methods available, which I will detail below. I agree with the authors that the native receptor population is an important and sometimes overlooked aspect of drug development, and I like that the discussion addresses the fact that different interacting proteins will be present in different cell types. The results gathered in this paper support their stance on their conclusions. The introduction has reviewed a good cross-section of the literature. Although the biosensors themselves are not novel, their use in neuronal cells is novel

and important. The receptors investigated also represent a good cross-section of druggable disease targets. The authors have also used a good variety of ligands to characterise the receptor G protein response.

We thank the referee for his/her very positive statements.

Questions and constructive feedback for the authors:

Major points

1) The major caveat that is not addressed throughout the whole paper is **the difference in GABA receptor expression between the two cell types** (CGNs and HEK cells). In line 164, the authors seem surprised that there is an increase in drug potency in the HEK cells. This is stated in lines 167-9 that neuronal receptors don't couple to Gi/o proteins as readily as HEK cells. Again, this is repeated in lines 181-3. In lines 190-2, the authors note that the constitutive response is not present in neurons – again this is probably due to a lower receptor density. Lines 335-7 fail to address the receptor density, although it is also true that a difference in cellular environment may cause changes in ligand effects. Reading the methods, it seems that the GABA receptors transfected into the HEK cells are under a CMV promoter in the pRK5 plasmid vector. This will cause receptor overexpression, which causes increase in potency due to the law of mass action (H J Motulsky and L C Mahan, Molecular Pharmacology 1984, 25 (1) 1-9). I suggest that the authors need to address this shortcoming by measuring receptor density in the two systems. Radioligand binding could be used for this, or, if a suitable antibody or fluorescent ligand is available, then ELISA, Western blotting and/or immunochemistry.

The referee is right. In the revised version, as asked by the referees #1 and #3, we analyzed the difference in receptor expression between the two cell types (CGNs and HEK) for the three receptors investigated by quantifying the amount of these receptors using membrane preparations by western blot (see new Fig. 1i, Fig. 4d and Fig. 5a). In these experiments same amounts of membranes from transfected HEK293 cells and neurons were analyzed. GABA_B receptors were five times more expressed in transfected HEK293 cells compared to neurons, CB1 receptor has a similar expression in the two systems. It is now better discussed in the revised version.

2) In the methods (lines 337-9) **it is not clear which species of receptors are being transfected into the recombinant HEK cell line**. In the primary neurons, the receptors will be of mouse origin, however in the cell lines this may be human – the NCBI database number for the receptor sequence is not given. I am not sure how conserved the receptors are between species, but this is an important point the authors need to address as differences in ligand potency with the biosensors could be due to a species difference in receptor.

The same point has been raised by referee #1. Our actual data were obtained using the mouse version of the CB1, α_{2A} receptors and the rat version of the GABA_B receptor, such that they can be compared with data obtained with the primary neurons prepared from mice. Of note the rat GABA_B receptor sequences differ by only two residues in GB1, and one residue in GB2 neither located close to the binding site nor within regions involved in the activation process.

3) Although the English is simple enough for a wide audience and the meanings are clear and concise on the whole, **the grammar is not always at a good standard**. For example, the first sentence (lines 40 and 41) reads: G protein-coupled receptors (GPCRs) ubiquitously expressed in every cell types, constitute the largest family of membrane proteins. This should be: G protein-coupled receptors (GPCRs) ARE ubiquitously expressed in every cell TYPE, AND constitute the largest family of membrane proteins, or, G protein-coupled receptors (GPCRs), ubiquitously expressed in every cell TYPE, constitute the largest family of membrane proteins. I will not point out all grammatical errors here, but **I suggest the authors get an external person to check the English grammar prior to submission in the future**.

The full manuscript has been revised by an English editor.

Minor points

4) In the abstract, GPCRs are classed as 'the most important drug targets' (line 27). Although I am a GPCR researcher and agree, I think researchers in other fields may disagree. Perhaps something more neutral, such as 'are the most targeted by currently registered drugs on the market'?

It has been changed.

5) I like that fact that all subtypes of Gi/o proteins were investigated (line 93). However, why was Gz omitted? This is another crucial member of the inhibitory G protein family.

The referee is right. Gz has now been included in the revised version of the study.

6) Lines 102-3 do not state the animal model which the neurons were taken from. This is important information for the reader.

It is from mouse. It is now clearly mentioned.

7) Lines 104-5 state that in vitro culturing of cerebellum granule neurons are 'a good model for the study of cellular and molecular mechanisms of survival and neuroprotection'. There is no reference to support this, or any reason given as to why this is the case, or why this is important to this paper.

It is now clearly indicated (ref 34).

8) Lines 115-20 I appreciate the comparison between the luciferases, however the authors may have missed a crucial reference where NanoLuciferase has already been inserted into these sensors previously: Schihada et al. 2021, Sci Signal DOI:10.1126/scisignal.abf1653, and/or Schihada et al. 2022, JBC DOI: 10.1016/j.jbc.2022.102328.

We agree. Dr Schulte's group did indeed publish similar sensors also using Nanoluc, but these studies were published after we wrote the first versions of our manuscript. The reference Schihada et al 2021 Sci Signaling has been mentioned both in the Introduction and Discussion of the revised manuscript.

9) Line 129 has an approximate measure of transfection of the cerebellar neurons. How was this counted? How can they be sure that all cells with the Venus tagged subunit also express the NanoLuc tagged alpha subunit? Did the authors consider a single plasmid transfection with a 2A cleavage site or IRES two-gene expression to ensure that cell transfection would produce both subunits?

Transfection efficiency was measured by the fluorescence of the transfected $VenusG\gamma_2$. It is now clarified in the revised version. We agree with the referee that it is possible that some cells express only $VenusG\gamma$ and not the $G\alpha^{Nluc}$ construct, although we think it probably not the case for many cells since the plasmids encoding the two different constructs were mixed before transfection. As suggested by the referee, we have tested a single plasmid transfection encoding both the $G\alpha_{i1}^{Nluc}$ and $VenusG\gamma$ constructs (for the three different $G\gamma$ subunits) with a T2A cleavage site between $G\alpha_{i1}^{Nluc}$ and the $VenusG\gamma$ constructs. But these constructs produced BRET sensors not functional (data not shown). We will use IRES two-gene expression in the next study.

10) Line 130 states that the low level of transfection is normal, which is true. However, it is not addressed that any transfected cell will be 'overexpressing' the sensors through the plasmid promoter.

This additional information has been added in the revised version.

11) Lines 138-9 states that the G proteins are 'barely activated' in CGNs compared to HEK

cells. This is a subjective statement. How much difference was there in potency and efficacy? Or were no suitable concentration response curves possible with the CGNs?

We agree. This sentence has been removed in the revised version. And concentration response curves in CGNs have been now performed for all the our Gi/o sensors (see new Supp Fig. 3e)

12) Lines 140-2 state that it is not the difference in sensor alpha subunits causing the difference seen between the two cell lines. Although this is true, I am wondering about the impact of beta subunit expression between the two systems? This is not controlled, as the native beta units are used in the sensor.

Yes, we agree that some of the differences can be due to the endogenous G β subunits, as also raised by referee #1 (point #6). This was part of our proposal that proteins that can be part of the RG complex can be different between neurons and HEK cells providing a possible explanation for the differences observed. We now specifically mention the G β subunits as a possibility. We also compared the expression of different endogenous G β subunits (except G β 4 since our antibody against this subunit is of poor quality) in both neurons and HEK (see new Supp Fig. 1a).

13) Lines 179-81 - I don't understand this sentence. Are the authors trying to say that Rac BHFF can cause G protein activation and calcium mobilization by itself in other systems? It is not a purely allosteric ligand?

We rephrased this sentence to clarify this point. In addition, we show additional results with the Rac BHFF used alone (concentration response curves both in CGNs and HEK293 cells) to show this allosteric ligand is an ago-PAM in transfected HEK, while its agonist activity is not detected in neurons (see new Supp 3j).

14) Do the authors mean 'expressed' rather than 'exhibited' in line 185?

Indeed, we replaced "exhibited" by "expressed".

15) Line 190 – do the authors mean that the inverse agonist is causing the receptor to be in its inactive conformation?

We clarified the sentence by adding the text "by stabilizing the inactive conformation of the receptor".

16) Lines 193-4 state that there is a difference in potency between the CGNs and HEK cells. Perhaps it should be addressed in the discussion that other endogenous receptors in the two cell types may impact these results.

This possibility was added in the Discussion of the revised manuscript.

17) Is the agonist Win 55,211-2, described in line 238, a super agonist?

To our knowledge, this compound was not reported as a super agonist.

18) I feel that the Emax does not need to be reported to two decimal places (lines 240-1). The span of the concentration response curve fit is probably not this accurate. One or no decimal places would be more precise.

The referee is right. It has been corrected.

19) Line 272 uses the word 'whatever'. This is used as a slang word in this context and is not suitable for a scientific paper. It is used again in line 354.

It has been corrected in the revised version.

20) Line 273 states that other sensors will be developed for the other G proteins. I refer back to the paper I pointed out where they have inserted the NanoLuc into many different G protein alpha subunits (Schihada et al. 2021, Sci Signal).

We have deleted this sentence in the revised version, but the reference Schihada et al. 2021, Sci Signal is now included both in the Introduction and Discussion.

21) In the discussion lines 312-3, it is stated that the poor coupling to Go proteins needs to be clarified. I would argue that this is not a problem as all receptors are different. I have found that some dopamine receptors prefer the Go subtype over Gi, and NPY receptors also (CNS expressed). Perhaps the receptors addressed in the paper are just not those associated with Go?

The referee is right. In the revised version we have deleted this paragraph. Indeed, as point out to the referees #1 and #3, we realized after submission of first version of our manuscript that two of our constructs (Gi3 and GoB) were indeed carrying a point mutation. Experiments were then repeated with validated constructs and the difference between the Go netBRET signals between HEK and neurons is smaller, but still clear for CB1 (Fig. 5 and Fig. 7).

REVIEWER COMMENTS

Reviewer #1 (Remarks to the Author):

The authors have made changes to address the previous concerns and the manuscript has improved. However, some of the critical points have not been adequately addressed because the authors have chosen to remove problematic or erroneous results, or acknowledge the problems in passing instead of definitely addressing the concerns with experimental evidence. Similarly, their conclusions have not changed significantly, as reflected, for example, by the fact that the title and abstract are essentially the same. I believe that the authors need to carry out additional experiments to substantiate some of the points that have been addressed only partially, and to make significant changes in their conclusions.

I ask the authors to please bear with me below, as I propose alternate text. It is not my intention to write your paper, but I provide the alternate text to be clear about what I mean because I think that in the last review cycle I might have been misunderstood.

1- The title says: "Specific pharmacological and Gi/o protein coupling properties of native GPCRs in neurons". This directly implies intrinsic properties of the GPCRs as the cause of the differences observed. However, the new results show that Gbeta subunit composition is completely different in the two systems investigated, which can easily explain the differences observed (as opposed to unique properties of the receptor in a given system). There is some discussion about this point, but the title does not reflect it. Based on this and other points developed below, something similar to "Different pharmacological and Gi/o protein specificity responses triggered by some endogenous GPCRs in neurons" would be more accurate. It is almost impossible to determine if the different responses observed are because of different properties of the receptor and/or G protein composition in the two systems.

Similarly, the statement in the discussion "we cannot rule out the influence of G β on G α subunit to the receptor coupling" seems insufficient. It is not only that you cannot rule it out, but also that it is very likely that the differences observed in the responses are caused by the major differences in Gbeta subunit composition.

2- Similarly, in the abstract I would replace "However, GPCR properties may differ in native environments because multiple factors can influence them" by "However, GPCR responses may differ in native environments because multiple factors can influence them". And "Here, by using bioluminescence resonance energy transfer Gi/o sensors, we revealed specific properties of native GPCRs in primary neurons" by "Here, by using bioluminescence resonance energy transfer Gi/o sensors, we revealed specific properties of responses triggered by native GPCRs in primary neurons"

3- The abstract also makes a point by saying "In addition, differences in agonist potencies in activating various G proteins were observed in neurons". This is after performing experiments in which for 2 of the 3 receptors investigated (GABA_B and α₂ adrenergic) showed levels of expression much higher in HEK293 cells than in neurons. This is something that needs to be completely addressed. The amount of receptor should be titrated down to levels similar to those observed in neurons and then assess the potency of ligands. This was started for the α₂ adrenergic receptor, but the authors did not reduce the levels of expression enough to be comparable to those in neurons (they still seem to be 10-15 times higher based on my calculations). It seems likely that differences in potency might disappear if the levels are equalized. This is important for the conclusions, because it would indicate that the pharmacological properties (agonist potency) are not a feature of the receptor in one context or another, but a consequence of experimental conditions that can be easily controlled in recombinant systems (just by reducing the amount of plasmid transfected). This is why saying "Some" in the title I proposed above would be justified (if equalizing expression levels negates the differences in responses across systems).

Depending on results, the authors will have to reword the abstract to accurately describe the findings (for example, "while agonist potencies were similar for some receptors (X and Y), some others (Z) displayed different potencies in cell lines versus neurons."

The discussion may also need to be revised "and in the potencies of some agonists (Figure 7b)"

4- Similarly, the authors should also titrate down the amount of receptor when investigating the allosteric modulator Rac B_{HFF}. It is not uncommon to find that "pure" PAMs behave as "ago-PAMs", as the authors observed when comparing GABA_B in neurons vs cell lines, when the receptor is overexpressed. Doing the comparison with similar levels of receptor expression will allow the authors to conclude if the receptor has the same or different properties.

Similarly, the measurements of constitutive GABA_B activity should be done by equalizing expression levels to those in neurons to determine if the observed differences with neurons are indeed related to expression levels. It is also not uncommon that receptor overexpression leads to constitutive activity.

5- The authors conclude "Our sensors revealed a ligand that seem biased as observed with the endogenous CB₁ receptor where the full agonist CP 55940 shows a much higher potency in activating Gα_{i1} compared to Gα_{oA} in neurons." As I mentioned in the original review, ligand bias claims should be in agreement with current guidelines in the field. Ligand bias has to be defined and quantified relative to a reference ligand in the same system. If the authors try to claim that the ligand can have different effect in different systems, that could be system bias or even a measurement bias. Maybe what the authors are trying to say is that the CP compound shows G_i vs G_o bias relative to one of the other compounds in neurons but not in cell lines. If so, the authors should *quantify* ligand bias for each G protein and system to determine if this is the case. It seems to me that CP is somewhat G_o biased relative to WIN in cell lines but not in neurons.

6- Minor: “a throughput analysis” and “a throughput format” are odd wordings. The throughput can be low, medium or high, but not an analysis or format in itself. If the authors meant “high throughput”, it is probably inadequate, as there is no demonstration of the high throughput capabilities (e.g., reproducibility scores). Maybe they meant that the assay can be done in a 96-well plate?

7- Minor: I hate to bring this up, but I think the text still needs some more revision for language. There are some grammar mistakes here and there, and the punctuation is odd in some places. Some examples:

“every cell types” should type

“expressed at a very low levels “ should be at very low levels or at a very low level

“the constructs were name named G α Nluc” should remove name

“Then to show the advantage of using Nluc over Rluc8, this latter was also inserted in the same position in G α i1 (named G α i1Rluc8; Figure 1d) and G α oA (named G α oARluc8; Supplementary Figure 1b) to measure BRET change upon agonist stimulation (see below).” It is odd in a couple places. I would just rewrite it.

“We have also validated our sensors for the analysis of the positive allosteric modulators (PAMs).” Should remove “the” before positive

“PAMs can synergistic enhance” should be synergistically

“We then used agonists to activate the other GPCRs expressed in the brain.” Should remove “the” before other

“Therefore, we have compared the results obtained above with VenusG γ 2 subunit with a sensor where the constructs G α Nluc were co-transfected with VenusG γ 8 or VenusG γ 9 well used in the G protein BRET sensors 25,36” I do not understand completely because of the “well used”

“depending of the receptor” should be depending on

“Regardless of the criticism of our approach, our study clearly reveal that data generated in recombinant systems should be taken with caution, before going into further pre-clinical and clinical development of drugs candidates only characterized this way” I would remove the comma after caution

And “higher potency than the other agonists in neurons (Figure 6b).” refers to the wrong fig

Reviewer #2 (Remarks to the Author):

This is an important paper on Class C GPCRs. It is mainly well done and the paper as written is interesting. I have just a couple of relatively minor concerns:

SPECIFIC CONCERNS:

1. For the endogenous lipid binding site the paper would be enhanced by docking and MD simulations to provide evidence in favor of the proposed lipid.
2. The paper has some minor grammatical errors which can be corrected at the proof stage.

Reviewer #3 (Remarks to the Author):

General comments:

The revised manuscript addressed some of the major concerns of the reviewers experimentally. Particular, the authors measured expression levels of receptors by means of western blot analysis. In addition, they performed interesting experiments with different Ggamma subtypes and found for some Galpha subunits clear differences in the agonist evoked BRET amplitude. Even though these data are now inserted into the manuscript, the conclusiveness of the results is somewhat limited and the original concerns about the origin of the differences reported for primary neurons and HEK cells have not vanished. Based on the new results, specifically the observation that the relative BRET changes for the different Galpha subtypes were clearly dependent on the Ggamma subunit I was rethinking their approach to compare delta BRET signals between different Galpha subunits. Based on these novel thoughts I have to include more concerns about the interpretation of the data and the novelty of the data compared to the initial review. I usually avoid to raise additional comments in the second review, however the novel results let to the following concern:

What does the absolute BRET change mean if the initial BRET values differ substantially between the different constructs?

The authors interpret the delta BRET signals as they simple reflect G protein subunit dissociation. However, two important issues are neglected: 1. The netBRET values in the absence of G protein activation are vastly different for the different Ga subtypes (and probably also Ggamma subtypes). There could be several reasons for that: differences in flexibility or exact positioning of the nLuc relative to

venus, differences in the complex formation capacity (competition with endogenous Galpha subunits etc. How much sense does it make then to compare the agonist change of the absolute BRET? Shouldn't it be at least normalized to the starting BRET signal?

2. It has been discussed that in intact cells both G protein subunits may not completely separate upon activation (ref23) a finding that may highly dependent on the G protein subtype (Frank et al. 2005). Even though there is good evidence from follow up papers, that the activated subunits actually loose affinity to each other and exchange faster (Digby et. al. 2006), there is no evidence that the subunits separate completely in intact cells. As reported for Ga13, effectors can influence both kinetics and direction of the RET signal between Galpha and Gbg subunits (Bodmann et al. 2017). Due to the high degree of uncertainty about the meaning of the maximal agonist BRET change for the selectivity of Receptor-G protein coupling, the concentration response curves are far more valuable. However, the main conclusions of the manuscript, that the pharmacological profile including the coupling profile of endogenous receptors show a major difference compared to HEK cells is supported only by a few examples where these differences are seen in a shift of the concentration response curves. As pointed out in the revised versions of the manuscript, the authors cannot exclude that these differences are due to differences in the expression levels of receptors and/or G proteins.

Specific Points:

In CGNs, we show that the BRET change induced by baclofen is dependent of the Gai/o
167 subunits, while these variations are less pronounced in HEK293 cells (Figure 1g and
168 Supplementary Figure 2c). This difference is not due to the level of expression of the Gai/o
169 protein subtypes, since a similar expression of these three Gai, two Gao and Gaz proteins
170 was measured in the CGNs (Figure 1h).

There is no statistical basis presented to support this statement.

We cannot exclude that the absence of

212 Rac BHFF agonist activity in CGNs is due to the lower GABAB receptor expression in
213 neurons compared to HEK293 cells (Figure 1i).

They write in 211-213,,We cannot exclude Why not writing: A possible explanation.....

Such discrimination between brimonidine and other agonists was not observed in HEK293
251 cells transfected with α 2A adrenoceptor (α 2AAR) (Figure 4b), even when the expression of
252 the α 2AAR was reduced (Supplementary Figure 4a-b). In addition, the data showed a
253 difference in the coupling properties of this receptor both in the transfected HEK293 cells and
254 of the endogenous receptor in CGNs (Figure 4c and Supplementary Figure 4c). Again, this
255 difference seems not due to the level of expression of the G α i/o protein subtypes, since a
256 similar expression of these three G α i, the two G α o and the G α z subtypes was measured in
257 each system (Supplementary Figure 4d). But it could be due to a major difference in the
258 level of expression of α 2AAR in the two systems as observed in western blot (Figure 4d).

With no proof that the alpha2A-AR receptor level is reduced to a comparable expression level than in neurons at the level of individual cells this finding is not of any real value.

The authors did a good job to convince the reader, that the expression of Ga and Gg did not alter the Gbeta expression

Reviewer #4 (Remarks to the Author):

I am satisfied with the authors' responses to my reviewer comments, all of which have been addressed, and congratulate the authors on their good work.

REVIEWER COMMENTS

Reviewer #1 (Remarks to the Author):

The authors have made changes to address the previous concerns and the manuscript has improved. However, some of the critical points have not been adequately addressed because the authors have chosen to remove problematic or erroneous results, or acknowledge the problems in passing instead of definitely addressing the concerns with experimental evidence. Similarly, their conclusions have not changed significantly, as reflected, for example, by the fact that the title and abstract are essentially the same. I believe that the authors need to carry out additional experiments to substantiate some of the points that have been addressed only partially, and to make significant changes in their conclusions.

I ask the authors to please bear with me below, as I propose alternate text. It is not my intention to write your paper, but I provide the alternate text to be clear about what I mean because I think that in the last review cycle I might have been misunderstood.

1- The title says: “Specific pharmacological and Gi/o protein coupling properties of native GPCRs in neurons”. This directly implies intrinsic properties of the GPCRs as the cause of the differences observed. However, the new results show that Gbeta subunit composition is completely different in the two systems investigated, which can easily explain the differences observed (as opposed to unique properties of the receptor in a given system). There is some discussion about this point, but the title does not reflect it. Based on this and other points developed below, something similar to “Different pharmacological and Gi/o protein specificity responses triggered by some endogenous GPCRs in neurons” would be more accurate. It is almost impossible to determine if the different responses observed are because of different properties of the receptor and/or G protein composition in the two systems.

We thank the reviewer for his comment and suggestion. We agree with the referee that the specific coupling properties may not be an intrinsic property of the receptor protein itself, but most probably the consequence of other proteins regulating the GPCR signaling. It is true that the expression "G protein coupling properties" may be interpreted as an intrinsic property of the receptor protein, though this was not our intention. To clarify this point, and as suggested by the referee, we propose a new title: “Specific pharmacological and Gi/o protein responses of some native GPCRs in neurons”. Meanwhile, we modified the abstract and added the information of the potential influence of G β and G γ composition to induce the G protein response difference.

Similarly, the statement in the discussion “we cannot rule out the influence of G $\beta\gamma$ on G α subunit to the receptor coupling” seems insufficient. It is not only that you cannot rule it out, but also that it is very likely that the differences observed in the responses are caused by the major differences in Gbeta subunit composition.

We agree with the reviewer. This part of the discussion (page 18) has been changed to better highlight that among the various possible explanations: “When analyzing the same G protein sensor, our study reveals major differences in G protein responses and agonist potencies between the neurons and heterologous HEK293 cells. One possibility can be the different composition of G β and G γ in CGNs and HEK293 cells. The difference in the profiles of G protein response using different G γ subunits (G γ 2, G γ 8 or G γ 9) (**Figure 6**) shows clearly the impact of G γ in CGNs. It shows that not only the G α subunit is important, but also the G β and G γ , consistent other recent studies (Masuho et al (2021) Cell Syst; Kankanamge et al (2022) JBC).”

2- Similarly, in the abstract I would replace “However, GPCR properties may differ in native environments because multiple factors can influence them” by “However, GPCR responses may differ in native environments because multiple factors can influence them”. And “Here, by using bioluminescence resonance energy transfer Gi/o sensors, we revealed specific properties of native GPCRs in primary neurons” by “Here, by using bioluminescence resonance energy transfer Gi/o sensors, we revealed specific properties of responses triggered by native GPCRs in primary neurons”

As suggested by the referee and to clarify that the functional differences observed are unlikely due to intrinsic properties of the receptor themselves, we modified the abstract as suggested by the referee:

- "However, GPCR responses may differ in native environments..."
- "...we revealed specific properties of Gi/o protein responses triggered by some native GPCRs in primary neurons".

3- The abstract also makes a point by saying "In addition, differences in agonist potencies in activating various G proteins were observed in neurons". This is after performing experiments in which for 2 of the 3 receptors investigated (GABA_B and α_2 adrenergic) showed levels of expression much higher in HEK293 cells than in neurons. This is something that needs to be completely addressed. The amount of receptor should be titrated down to levels similar to those observed in neurons and then assess the potency of ligands. This was started for the α_2 adrenergic receptor, but the authors did not reduce the levels of expression enough to be comparable to those in neurons (they still seem to be 10-15 times higher based on my calculations). It seems likely that differences in potency might disappear if the levels are equalized. This is important for the conclusions, because it would indicate that the pharmacological properties (agonist potency) are not a feature of the receptor in one context or another, but a consequence of experimental conditions that can be easily controlled in recombinant systems (just by reducing the amount of plasmid transfected). This is why saying "Some" in the title I proposed above would be justified (if equalizing expression levels negates the differences in responses across systems).

Depending on results, the authors will have to reword the abstract to accurately describe the findings (for example, "while agonist potencies were similar for some receptors (X and Y), some others (Z) displayed different potencies in cell lines versus neurons."

The discussion may also need to be revised "and in the potencies of some agonists (Figure 7b)"

According to the reviewer's suggestion, we examined the expression level and functional properties of both GABA_B and α_2 adrenergic receptors by varying the amount of plasmids encoding these receptors (keeping the total amount of cDNA constant with empty vectors) used for transfection, in HEK cells and compared them with those measured in CGNs.

As shown in the newly added results, conditions in which the expression level of GABA_B receptors was reduced reaching a similar level as that in CGNs (new Figure 1h), were identified (GB1 and GB2 encoding plasmids: 9 ng + 12 ng VS previous amount 30 ng + 40 ng per well in 96-well plate), and G protein sensor responses are now measured in these conditions (new Figure 1g). In addition, under such conditions, the baclofen pEC50 was similar to that observed with higher expression levels, and still significantly different from that measured in CGNs (new Figure 2g). Meanwhile, the Rac BHFF still showed agonist activity in transfected HEK293 cells with GABA_B receptor expression similar to that found in CGNs where no agonist activity of this PAM could be detected (Figure 2h). These results are consistent with our previous interpretation.

For the α_2 adrenergic receptor, we also identified a condition in which a similar expression level as that of CGNs could be observed when using 15 ng of encoding plasmid per well (new Figure 4c). G protein sensor responses are now measured in these conditions (new Figure 4e). In addition, under such conditions, the brimonidine pEC50 was similar to that observed with higher expression levels, and still significantly different from that measured in CGNs (new Figure 4d). As we already analyzed the functional properties of this receptor at 20 ng and 10 ng transfection conditions in our previous Supplementary Figure 4a, we moved this figure to the main Figure 4b. As well illustrated in this figure, the α_2 adrenergic receptor agonists showed full activation in all these transfection amounts (10 ng, 20 ng, 50 ng) in HEK293 cells.

The description has been re-worded in the abstract as "we revealed specific properties of G_{i/o} protein-mediated responses triggered by GABA_B, α_2 adrenergic and cannabinoid CB1 receptors in primary neurons, different from those in heterologous cells. These include different profiles in the G_{i/o} protein subtypes-mediated responses, and differences in the potencies of some ligands even at similar receptor expression levels."

4- Similarly, the authors should also titrate down the amount of receptor when investigating the allosteric modulator Rac BHFF. It is not uncommon to find that "pure" PAMs behave as "ago-PAMs", as the authors observed when comparing GABA_B in neurons vs cell lines, when the receptor is overexpressed. Doing the

comparison with similar levels of receptor expression will allow the authors to conclude if the receptor has the same or different properties.

Similarly, the measurements of constitutive GABA_B activity should be done by equalizing expression levels to those in neurons to determine if the observed differences with neurons are indeed related to expression levels. It is also not uncommon that receptor overexpression leads to constitutive activity.

This point has also been taken into consideration as indicated in our response to the previous point. Although the referee is right that over-expression might increase the measured proportion of constitutive activity, due for example to a high proportion of receptor reserve, but this is not the only possible explanation. As we already reported for the mGlu1/5 receptors, their constitutive activity can be regulated by the intracellular homer proteins (Ango et al., Nature 2001; Bockaert et al Ango J. Neurosci 2021), unrelated to their expression level.

5- The authors conclude “Our sensors revealed a ligand that seem biased as observed with the endogenous CB1 receptor where the full agonist CP 55940 shows a much higher potency in activating G α i1 compared to G α oA in neurons.” As I mentioned in the original review, ligand bias claims should be in agreement with current guidelines in the field. Ligand bias has to be defined and quantified relative to a reference ligand in the same system. If the authors try to claim that the ligand can have different effect in different systems, that could be system bias or even a measurement bias. Maybe what the authors are trying to say is that the CP compound shows Gi vs Go bias relative to one of the other compounds in neurons but not in cell lines. If so, the authors should *quantify* ligand bias for each G protein and system to determine if this is the case. It seems to me that CP is somewhat Go biased relative to WIN in cell lines but not in neurons.

Yes, the referee is perfectly correct, and this was a mistake in our manuscript. This has been corrected and we now simply report the difference observed without referring to ligand bias property. We discuss the possible reasons for such a difference, including ligand bias signaling, or regulation by intracellular partners. We corrected the description in the Discussion as:

"Our sensors revealed that the CB1 agonist, CP 55940 has a higher potency in generating G α i1 than G α oA responses in neurons, relative to the other agonists tested. An additional characterization of this ligand with methods that do not need a transfected sensor would be necessary to confirm a possible bias in signaling. Indeed, such biased signaling between Gi and Go proteins have already been reported upon agonist stimulation^{12,13,74}, by allosteric modulators⁴⁹ or as a result of genetic variation⁷⁵. In our study, the molecular bases of this difference in agonist effect is unclear. CP 55940 and Win 55212-2 (K_i 1.1 nM and 62.3 nM for CB1 receptor, respectively)^{76,77} have different scaffolds. Their differential effects might be explained by the ligand-binding kinetics⁷⁸, by specific active conformation stabilized by the agonist⁷⁵, or due to specific components associated to the receptor in these neurons, better stabilizing the CP 55940/CB1/Gi complex."

6- Minor: “a throughput analysis” and “a throughput format” are odd wordings. The throughput can be low, medium or high, but not an analysis or format in itself. If the authors meant “high throughput”, it is probably inadequate, as there is no demonstration of the high throughput capabilities (e.g., reproducibility scores). Maybe they meant that the assay can be done in a 96-well plate?

We thank the reviewer for this comment. The sentence has been modified as suggested “These sensors allow a simple analysis of different types of GPCR ligands on each G α _{i/o} protein subtypes in primary neurons in a medium throughput format (in 96-well plates)”.

7- Minor: I hate to bring this up, but I think the text still needs some more revision for language. There are some grammar mistakes here and there, and the punctuation is odd in some places. Some examples:

“every cell types” should type

“expressed at a very low levels “ should be at very low levels or at a very low level

“the constructs were name named G α Nluc” should remove name

“Then to show the advantage of using Nluc over Rluc8, this latter was also inserted in the same position in G α i1 (named G α i1Rluc8; Figure 1d) and G α oA (named G α oARluc8; Supplementary Figure 1b) to measure BRET change upon agonist stimulation (see below).” It is odd in a couple places. I would just rewrite it.

“We have also validated our sensors for the analysis of the positive allosteric modulators (PAMs).” Should remove “the” before positive

“PAMs can synergistic enhance” should be synergistically

“We then used agonists to activate the other GPCRs expressed in the brain.” Should remove “the” before other

“Therefore, we have compared the results obtained above with VenusG γ 2 subunit with a sensor where the constructs GaNluc were co-transfected with VenusG γ 8 or VenusG γ 9 well used in the G protein BRET sensors 25,36” I do not understand completely because of the “well used”

“depending of the receptor” should be depending on

“Regardless of the criticism of our approach, our study clearly reveal that data generated in recombinant systems should be taken with caution, before going into further pre-clinical and clinical development of drugs candidates only characterized this way” I would remove the comma after caution

We thank the reviewer for pointing these mistakes. We corrected all of them in the revised version and the manuscript has been further checked again by a language English editor.

And “higher potency than the other agonists in neurons (Figure 6b).” refers to the wrong fig

We have corrected this mistake in the revised version.

Reviewer #2

Their concerns have been fully addressed.

We thank the reviewer for this positive comment.

Reviewer #3 (Remarks to the Author):

General comments:

The revised manuscript addressed some of the major concerns of the reviewers experimentally. Particular, the authors measured expression levels of receptors by means of western blot analysis. In addition, they performed interesting experiments with different Ggamma subtypes and found for some Galpha subunits clear differences in the agonist evoked BRET amplitude. Even though these data are now inserted into the manuscript, the conclusiveness of the results is somewhat limited and the original concerns about the origin of the differences reported for primary neurons and HEK cells have not vanished. Based on the new results, specifically the observation that the relative BRET changes for the different Galpha subtypes were clearly dependent on the Ggamma subunit. I was rethinking their approach to compare delta BRET signals between different Galpha subunits. Based on these novel thoughts I have to include more concerns about the interpretation of the data and the novelty of the data compared to the initial review. I usually avoid to raise additional comments in the second review, however the novel results let to the following concern:

What does the absolute BRET change mean if the initial BRET values differ substantially between the different constructs?

The authors interpret the delta BRET signals as they simple reflect G protein subunit dissociation. However, two important issues are neglected:

1. The net BRET values in the absence of G protein activation are vastly different for the different Ga subtypes (and probably also Ggamma subtypes). There could be several reasons for that: differences in flexibility or exact positioning of the nLuc relative to venus, differences in the complex formation capacity (competition with endogenous Galpha subunits etc. How much sense does it make then to compare the agonist change of the absolute BRET? Shouldn't it be at least normalized to the starting BRET signal?

This is an excellent point, as we fully agree that each BRET sensor has its own properties, with its specific range of signal, even though they were constructed similarly. In addition, differences may also come from the preferential association with specific G $\beta\gamma$ (Masuho et al (2021) Cell Syst). As such, we agree that it is

not possible to compare the ligand-induced change in BRET signal between sensors. However, we think it is acceptable to compare the signals obtained with the same sensor in different cell types, with the limit of the possible influence of the endogenous G proteins. However, the later are very likely far less expressed than the transfected sensor.

Moreover, it is true that basal signal measured in the absence of agonist, should be taken into account. As such, we decided to change most of our figures in two ways:

- 1) The signal is now expressed as the agonist-induced change in BRET ratio expressed as percentage of the basal signal ($(\text{BRET}_{\text{basal}} - \text{BRET}_{\text{ago}} / \text{BRET}_{\text{basal}}) \times 100$).
- 2) We now systematically compare the change in BRET of a given sensor in HEK cells and CGNs.

We agree that such a representation is more accurate. Of importance, even taking all these into consideration, this does not change our conclusion regarding the observed differences in response of the native GPCRs tested in neurons, and the transfected ones in HEK cells.

Additionally, the difference in the profiles of G protein response using different $G\gamma$ subunit in CGNs (Figure 6c) suggested that not only the $G\alpha$ subunit is important, but also the $G\beta$ and $G\gamma$, consistent with the concept in other reports (Masuho et al (2021) Cell Syst; Kankanamge et al (2022) JBC).

2. It has been discussed that in intact cells both G protein subunits may not completely separate upon activation (ref23) a finding that may highly dependent on the G protein subtype (Frank et al. 2005). Even though there is good evidence from follow up papers, that the activated subunits actually loose affinity to each other and exchange faster (Digby et. al. 2006), there is no evidence that the subunits separate completely in intact cells. As reported for $G\alpha_{13}$, effectors can influence both kinetics and direction of the RET signal between $G\alpha$ and $G\beta\gamma$ subunits (Bodmann et al. 2017). Due to the high degree of uncertainty about the meaning of the maximal agonist BRET change for the selectivity of Receptor-G protein coupling, the concentration response curves are far more valuable. However, the main conclusions of the manuscript, that the pharmacological profile including the coupling profile of endogenous receptors show a major difference compared to HEK cells is supported only by a few examples where these differences are seen in a shift of the concentration response curves. As pointed out in the revised versions of the manuscript, the authors cannot exclude that these differences are due to differences in the expression levels of receptors and/or G proteins.

We thank the reviewer for this comment. We fully agree that G protein sensor response can result either from a complete dissociation between $G\alpha$ and $G\beta\gamma$, but can also result from a repositioning of $G\beta\gamma$ relative to $G\alpha$ without a real dissociation. This is now clearly stated in our revised manuscript (Page 17).

Regarding the second point of the referee about the receptor expression level, this was also one of the main point of the first referee. As indicated in our detailed answer to the 1st referee, the properties observed in the transfected cells, were still different from those in CGNs when analyzed at a similar receptor expression level (new Figure 1g, new Figure 2g-h and new Figure 4d-e). The potency of GABA_B agonist baclofen (new Figure 2g) and bromonidine agonist for α_2 adrenergic receptors (new Figure 4d) did not change in the reduced expression level. So we think it is the difference between two cell systems, and in addition to receptor expression level, many other factors can influence the potency.

Specific Points:

*166 In CGNs, we show that the BRET change induced by baclofen is dependent of the Gai/o
167 subunits, while these variations are less pronounced in HEK293 cells (Figure 1g and
168 Supplementary Figure 2c). This difference is not due to the level of expression of the Gai/o
169 protein subtypes, since a similar expression of these three Gai, two Gao and Gaz proteins
170 was measured in the CGNs (Figure 1h).*

There is no statistical basis presented to support this statement.

As indicated above, we agree it might be less critical to compare the difference in signal generated by the same sensor in different cells, but rather the difference in signal generated by different sensors generated in the same cells. This paragraph has then been modified accordingly:

“We then compared the agonist-induced BRET signals obtained with the same $G\alpha_{i/o}$ sensor in the two different cell types, CGNs and HEK293 cells (**Figure 1g**). Conditions were optimized such that the $GABA_B$ receptor expression level in HEK293 cells was similar to that in CGNs (**Figure 1h**). Since the basal BRET signal in the absence of agonist can be different between the G protein sensors (**Supplementary Figure 2c**), the signal was expressed as the agonist-induced change in BRET ratio expressed as percentage of the basal signal. The results showed that in CGNs, the BRET changes induced by baclofen are significantly different from those measured in HEK293 cells except for the two G_o sensors (**Figure 1g**). Indeed, a larger BRET signal is measured in CGNs with G_{i1} and G_{i3} , and a large signal is observed with G_z in HEK293 cells while absent in CGNs. The differences observed are related neither to the expression levels of the $G\alpha^{Nluc}$ sensor components (**Figure 1i**) nor to the endogenous $G\alpha_{i/o}$ mRNA expression levels (**Figure 1j**). Though lower signal was observed in G_{i2} in CGNs, the difference of baclofen response can be detected in dose-dependent manner (**Supplementary Figure 2d**). Similar pEC_{50} values were measured in three G_i subtype sensors and two G_o subtype sensors in CGNs upon baclofen activation (**Supplementary Figure 2d**).”

211 We cannot exclude that the absence of
212 Rac BHF agonist activity in CGNs is due to the lower $GABA_B$ receptor expression in
213 neurons compared to HEK293 cells (Figure 1i).

They write in 211-213,,We cannot exclude Why not writing: A possible explanation.....
This sentence has been changed as “A possible explanation” in the revised version.

250 Such discrimination between brimonidine and other agonists was not observed in HEK293
251 cells transfected with α_2AAR (Figure 4b), even when the expression of
252 the α_2AAR was reduced (Supplementary Figure 4a-b). In addition, the data showed a
253 difference in the coupling properties of this receptor both in the transfected HEK293 cells and
254 of the endogenous receptor in CGNs (Figure 4c and Supplementary Figure 4c). Again, this
255 difference seems not due to the level of expression of the Gai/o protein subtypes, since a
256 similar expression of these three Gai , the two Gao and the Gaz subtypes was measured in
257 each system (Supplementary Figure 4d). But it could be due to a major difference in the
258 level of expression of α_2AAR in the two systems as observed in western blot (Figure 4d).

With no proof that the α_2AAR receptor level is reduced to a comparable expression level than in neurons at the level of individual cells this finding is not of any real value.

We detected the membrane expression of α_2AR with these reduced transfection amount (10 ng, 20 ng and also 15 ng per well). And with the transfection of 15 ng, the membrane expression of α_2AR receptor is similar as that in CGNs. These data are shown in the new Figure 4d. As we already detected the 20 ng and 10 ng conditions in previous Supplementary Figure 5b (now Figure 4b), and the α_2AR agonists in HEK293 cells showed full activation in all these transfection amounts (10 ng, 20 ng, 50 ng).

The authors did a good job to convince the reader, that the expression of Ga and Gg did not alter the $Gbeta$ expression

We thank the reviewer for this positive comment.

Reviewer #4 (Remarks to the Author):

I am satisfied with the authors' responses to my reviewer comments, all of which have been addressed, and congratulate the authors on their good work.

We thank the reviewer for this positive comment.

REVIEWERS' COMMENTS

Reviewer #1 (Remarks to the Author):

The authors have again improved the manuscript by addressing many of the points previously raised. I do not take pleasure in prolonging this process, but I think that there are two points that require further clarification (no experimental work).

1- A point raised earlier was about the influence of Gbg composition on the responses detected. This has been partially addressed, but based on the results presented the authors should explicitly acknowledge that a limitation of the biosensors described is that what is detected is significantly influenced by the components of the biosensor expressed. I am referring to the results showing clear response differences depending on the tagged Ggamma subtype used, which make evident that what is detected depends on the biosensor components used rather than on properties intrinsic to the system under study.

2- When addressing previous comments on the pharmacological properties of cannabinoid receptors, the authors changed the text to the following: "Our sensors revealed that the CB1 agonist CP 55940 has a higher potency in generating Gi1 than GoA responses in neurons, relative to the other agonists tested. An additional characterization of this ligand with methods that do not need a transfected sensor would be necessary to confirm a possible bias in signaling."

However, it is not correct that one would need methods that do not involve a biosensor. As long as the method of detection is consistent across ligands in the same cell system and the authors define the reference ligand, they could quantify bias as suggested in the previous round of reviews. Moreover, the new text changes does not make clear if the authors are trying to make a point about the differences observed to be something unique to neurons that does not happen in cell lines (eg, is the higher potency for Gi over Go something that is observed only in neurons?), which seemed to be the original intention. The latter could be done upon quantification of bias factors.

Reviewer #3 (Remarks to the Author):

The revised manuscript address all the concerns I raised in my previous review. I particularly liked the inclusion of the data on the low receptor expressing HEK cells, which are now more comparable to the situation in neurons. Very much improved manuscript!

A point-by-point response to the reviewers' comments

Reviewer #1 (Remarks to the Author):

The authors have again improved the manuscript by addressing many of the points previously raised. I do not take pleasure in prolonging this process, but I think that there are two points that require further clarification (no experimental work).

1- A point raised earlier was about the influence of Gbg composition on the responses detected. This has been partially addressed, but based on the results presented the authors should explicitly acknowledge that a limitation of the biosensors described is that what is detected is significantly influenced by the components of the biosensor expressed. I am referring to the results showing clear response differences depending on the tagged Ggamma subtype used, which make evident that what is detected depends on the biosensor components used rather than on properties intrinsic to the system under study.

We thank the reviewer for this comment. The description of the data obtained with the various Ggamma subunit and the limitation of the biosensor has been added in the discussion: "Therefore, it brings to our attention that the G protein-mediated response also depends on the biosensor components used, which is the limitation of biosensors that only detect the response of a well-defined G protein. Meanwhile, the difference can be due to the intrinsic properties of the system under study."

We therefore draw attention to the fact that the G protein response detected also depends on the biosensor components used, which is the limitation of biosensors that only detect the response of a well-defined G protein. In addition, the difference may be due to the intrinsic properties of the system under study that only detect the response of a well-defined G protein.

2- When addressing previous comments on the pharmacological properties of cannabinoid receptors, the authors changed the text to the following: "Our sensors revealed that the CB1 agonist CP 55940 has a higher potency in generating Gi1 than GoA responses in neurons, relative to the other agonists tested. An additional characterization of this ligand with methods that do not need a transfected sensor would be necessary to confirm a possible bias in signaling."

However, it is not correct that one would need methods that do not involve a biosensor. As long as the method of detection is consistent across ligands in the same cell system and the authors define the reference ligand, they could quantify bias as suggested in the previous round of reviews. Moreover, the new text changes does not make clear if the authors are trying to make a point about the differences observed to be something unique to neurons that does not happen in cell lines (eg, is the higher potency for Gi over Go something that is observed only in neurons?), which seemed to be the original intention. The latter could be done upon quantification of bias factors.

We thank the reviewer to point this out. As requested, we analyzed the bias factor using the ligand Win 55,212-2 as a reference, following the guideline (Community guidelines for GPCR ligand bias: IUPHAR review 32, Br J Pharmacol. 2022 Jul;179(14):3651-3674). A new supplementary Table 3

reporting this analysis was added in the supplementary file.

The indicated description was added in the result section, page 8 as “when taking Win 55,212-2 as a reference ligand, Bay 59-3074 showed G_{oA} bias in both CGNs and HEK293 cells with a bias factor of 1.6 and 1.3 respectively. CP 55940 showed a G_{i1} bias in CGNs and a G_{oA} bias in HEK293 cells with a bias factor of 1.6 and 1.4 respectively, indicating a difference between G_{i1} and G_{oA} proteins revealed by CP 55940”. The discussion section has also been modified in page 11 as “Our sensors showed a difference between G_i and G_o proteins revealed by a CB1 agonist CP 55940, which has a higher potency in generating G_{i1} than G_{oA} responses in neurons, relative to the other agonists tested”.

Reviewer #3 (Remarks to the Author):

The revised manuscript addresses all the concerns I raised in my previous review. I particularly liked the inclusion of the data on the low receptor expressing HEK cells, which are now more comparable to the situation in neurons. Very much improved manuscript!

We thank the reviewer for this very positive comment.